# Associations between components of household expenditures and the rate of change in the number of new confirmed cases of COVID-19 in Japan: Time-series analysis

Hajime Tomura ®*

Faculty of Political Science and Economics, Waseda University, Shinjyuku, Tokyo, Japan

* htomura@waseda.jp

## Abstract

**Data Availability Statement:** https://github.com/hajimetomura/COVID19_household_exp.

**Funding:** The author received no specific funding for this work.

### Background

Social distancing measures to prevent the spread of COVID-19 included restrictions on retail services in many countries. In some countries, the governments also subsidized consumer spending on part of retail services to help struggling businesses. To evaluate the costs and benefits of government interventions in retail services, it is necessary to measure the infectiousness of each type of consumer activity.

### Methods

This study regresses the log difference over seven days in the number of new confirmed cases of COVID-19 in Japan on lagged values of household expenditures per household on eating out, traveling, admissions to entertainment facilities, clothing and footwear, and the other items, as well as a measure of mobility in public transportation in the past 14 days. The sample period of the dependent variable is set from March 1, 2020, to February 1, 2021, in order to avoid a possible structural break due to the spread of mutant strains in 2021. The regression model is estimated by the Bayesian method with a non-informative (improper) prior. The estimated model is evaluated by out-of-sample forecast performance from February 2, 2021, onward.

### Results

The out-of-sample forecasts of the regression by the posterior means of regression coefficients perform well before the spread of the Delta variant in Japan since June 2021. $R^2$ for the out-of-sample forecasts from February 2, 2021, to June 30, 2021, is 0.60. The dependent variable of the regression overshot the out-of-sample forecasts from mid-June to August 2021. Then, the out-of-sample forecasts overpredicted the dependent variable for the rest of 2021.

**Competing interests:** The author have declared that no competing interests exist.

## Conclusion

The estimated model can be potentially useful in simulating changes in the number of new confirmed cases due to household spending on retail services, if it can be adjusted to real-time developments of mutant strains and vaccinations. Such simulations would help in designing cost-efficient government interventions.

## Introduction

After the onset of the COVID-19 pandemic, social distancing measures were introduced as part of non-pharmaceutical interventions (NPIs) to contain the spread of COVID-19 across countries [1]. The mitigating effects of these measures on the transmission of COVID-19 have been confirmed by the estimates of associations between the introduction of NPIs and subsequent reductions in the time-varying reproduction number, i.e., the average number of secondary cases of infection generated by a primary case at each point of time, across countries [2–10]. Similar associations have been also found between NPIs and the number of reported cases of COVID-19 or the COVID-related mortality rate [11, 12]. These findings are consistent with the prediction of simulation analysis using theoretical epidemiological models [13–15].

Social distancing measures, however, caused revenue losses for businesses, as they restricted physical retail services by stay-at-home orders for residents or more targeted interventions in specific businesses, such as temporary closures of bars and restaurants [16]. Thus, there occurred a trade-off between containing the spread of COVID-19 and maintaining economic activities [17]. This trade-off has become a major challenge for policy makers across countries.

To compare the costs and benefits of social distancing measures, it is necessary to associate the protective effects of these measures with the economic values of restricted activities. For this agenda, hybrid frameworks have been developed to combine epidemiological models with economic models in which the effects of NPIs on economic production are calibrated [18, 19]. While there is no feedback effect from economic activities to the spread of COVID-19 in this type of framework, there is also a different type of model developed by theoretical economists to introduce behavioral responses by households and firms to COVID-19 and NPIs into epidemiological models [20–22]. This type of model features inter-dependence between aggregate (i.e., total) consumption in the economy and the spread of COVID-19, focusing on the responses of the entire economy to COVID-19.

One of the remaining questions is to estimate associations between the values of more detailed types of consumer activities and the spread of COVID-19. This question is motivated by the fact that governments tended to intervene in only limited ranges of retail services, in order to minimize associated revenue losses for businesses [16, 23]. Also, in some countries, governments subsidized consumer spending on part of retail services to help struggling businesses. For example, the U.K. government introduced the "Eat Out to Help Out Scheme" in August 2020, which offered 50% discounts on meals up to 10 pounds per diner after the nationwide lockdown from March to May 2020 [24]. Also, the Japanese government introduced the "Go-To-Eat" and "Go-To-Travel" campaigns in the second half of 2020, which subsidized the expenses of eating out, including alcoholic drinks, and the expenses of accommodation and transportation for domestic travels, respectively [25, 26]. The introduction of these campaigns followed the declaration of the first state of emergency from April to May 2020 by the Japanese government.

On the Eat Out to Help Out Scheme in the U.K., an association between the number of bars and restaurants participating in the scheme and the spread of COVID-19 in each region has been confirmed by a panel study using rainfalls for exogenous instruments in the difference-in-differences method [27]. On the Go-To-Travel campaign in Japan, an association between participation in the campaign and the incidence of symptoms indicative of COVID-19 has been reported by a cross-sectional study using Internet survey data [28]. More generally, there have been studies estimating associations between people's visits to retail services and the spread of COVID-19 by using mobility data across countries [29–36]. While these studies confirmed associations between detailed types of consumer activities and the spread of COVID-19, they utilized data on physical visits or participation for the measures of consumer activities. It remains an issue to associate the economic values of detailed types of consumer activities with their contributions to the spread of COVID-19.

On this issue, this study estimates associations between classified components of household expenditures and the rate of change in the number of new confirmed cases of COVID-19 by applying a time-series regression model to daily nationwide data in Japan. The estimated regression coefficients measure the infectiousness of each type of consumer activity per value of spending per household. In addition, household expenditure data are also convenient to estimate associations between detailed types of consumer activities and the spread of COVID-19 simultaneously, because they are categorized according to the classification of goods and services purchased by households. Using the estimated regression coefficients, this study decomposes the rate of change in the number of new confirmed cases of COVID-19 on each date into its associations with the values of spending on classified categories of consumer goods and services, including eating out and traveling.

## Methods

### Regression model

This study regresses the log difference over seven days in the number of new confirmed cases of COVID-19 on lagged values of classified components of household expenditures and a measure of mobility in public transportation in the past 14 days. The regression form is derived from the law of motion for the number of new infections described by [37]:

$$I_t = R_t \sum_{s=1}^{t} I_{t-s} w_s + \text{error}, \tag{1}$$

where $I_t$ is the number of new infections on date $t$, $R_t$ is the instantaneous reproduction number on date $t$, and $w_s$ is the infectivity function for infection incidence $s$ days ago. The first term on the right-hand side is the expected value of $I_t$, given the past numbers of new infections, $\{I_{t-s}\}_{s=1}^{t}$. The second term is expectation error due to the difference between the expected value and the realized value of $I_t$.

To derive the regression form for this study, a formula for the number of new confirmed cases of infection on date $t$, $N_t$, is also specified as follows:

$$N_t = \sum_{m=1}^{M} I_{t-m}(1-\mu)p_m + \text{error}, \tag{2}$$

where the first term on the right-hand side is the number of new symptomatic patients on date $t$, for which $p_m$ denotes the probability of the incubation period being $m$ days, $M$ is the maximum length of an incubation period, and $\mu$ is the probability that an infected individual is asymptomatic. These probabilities are multiplied to the number of new infections $m$ days ago,

$I_{t-m}$, assuming that the law of large numbers is applicable. The second term is the residual accounting for reporting delays, false test results, and the detection of presymptomatic and asymptomatic infected individuals by testing.

Combining Eqs (1) and (2) yields the following equation:

$$N_t = \sum_{m=1}^{M} R_{t-m} \sum_{s=1}^{t-m} I_{t-m-s} w_s (1 - \mu) p_m + \text{error}, \tag{3}$$

where the second term on the right-hand side is the residual due to the error terms in Eqs (1) and (2). Because the number of new infections, $I_t$, is not directly observable, the total infectiousness of infected individuals, $\sum_{s=1}^{t-m} I_{t-m-s} w_s$, for each incubation period, $m$, is approximated by the number of new confirmed cases in the middle of the range of incubation periods, $N_{t-M/2}$, with a linear coefficient, $\phi$:

$$\sum_{s=1}^{t-m} I_{t-m-s} w_s \approx \phi N_{t-M/2} \quad \text{for } m = 1, 2, ..., M, \tag{4}$$

Because the total infectiousness of infected individuals is summed over the range of incubation periods on the right-hand side of Eqs (3) and (4) aims to approximate the average infectiousness of infected individuals over the range of incubation periods by the number of new confirmed cases of infection in the middle of the range. If Eq (4) is substituted into Eq (3), it yields the following equation:

$$\frac{N_t}{N_{t-M/2}} = \sum_{m=1}^{M} R_{t-m} \phi (1 - \mu) p_m + \text{error} \tag{5}$$

where the second term on the right-hand side is the residual due to the error term in Eq (3) and the approximation error in Eq (4).

The instantaneous reproduction number, $R_t$, is assumed to be the following function of explanatory variables:

$$R_t \phi (1 - \mu) = \alpha_0 + \alpha_1 D_{NY,t} + \alpha_2 D_{AH,t} + \sum_{i=1}^{I} \beta_i D_{SoE,i,t}$$
$$+ \sum_{j=1}^{J} \left( \gamma_j X_{j,t} + \theta_j D_{AH,t} X_{j,t} + \sum_{i=1}^{I} \psi_{ij} D_{SoE,i,t} X_{j,t} \right), \tag{6}$$

where $D_{NY,t}$ is a time dummy for the year-end and new-year holiday period in Japan, $D_{AH,t}$ is a dummy variable for absolute humidity, $D_{SoE,i,t}$ for $i = 1, 2, ..., I$ is a time dummy for each state of emergency or the period before the first state of emergency, $I$ is the number of these time dummies related to states of emergency, $\{X_{j,t}\}_{j=1}^{J}$ is a set of classified components of household expenditures and a measure of mobility in public transportation, and $J$ is the number of explanatory variables in this set.

The underlying assumption for Eq (6) is that each person has more chances of physical contacts with other people if there are more consumer activities in the country. Because consumer activities associated with the same category of household expenditures can have different degrees of infectiousness depending upon the circumstances of spending, such as whether they are online or offline purchases, the coefficient of each variable in Eq (6) captures the average effect of the variable. Mobility in public transportation is also included as a separate explanatory variable, because the infectiousness of physical contacts in public transportation is

unlikely to depend on what passengers do at their destinations. Thus, this effect should be measured separately from the infectiousness of consumer activities. In contrast, a measure of mobility at the places of retail services is not included in the explanatory variables, because the infectiousness of each type of consumer activity is captured by each component of household expenditures among the explanatory variables.

In addition, there are cross terms on the right-hand side of Eq (6), because the infectiousness of consumer activities and mobility in public transportation may depend on the weather, and might vary across each state of emergency and the transition period between the onset of the pandemic and the first state of emergency. In addition, the year-end and new-year holiday dummy is included in the explanatory variables because it is customary for people to have home parties with families and friends, which can be infectious, during the holiday period around the new year's day in Japan. The gatherings at home are not retail services, and thus difficult to capture with household expenditure data.

Because the gross rate of change in the number of new confirmed cases on the left-hand side of Eq (5) cannot be negative by definition, it is replaced with the log difference in the number of new confirmed cases, $\ln N_t - \ln N_{t-M/2}$, for the dependent variable in Eq (5), so that there is no need for a restriction on the probability distribution of residuals in the regression due to a non-negativity constraint on the dependent variable. Then, substituting Eq (6) into Eq (5) yields the final form of the regression:

$$\ln N_t - \ln N_{t-M/2} = \alpha_0 + \alpha_1 F(D_{NY,t}) + \alpha_2 F(D_{AH,t}) + \sum_{i=1}^{I} \beta_i F(D_{SoE,i,t})$$
$$+ \sum_{j=1}^{J} \left( \gamma_j F(X_{j,t}) + \theta_j F(D_{AH,t} X_{j,t}) + \sum_{i=1}^{I} \psi_{ij} F(D_{SoE,i,t} X_{j,t}) \right) + \epsilon_t, \tag{7}$$

where $F$ denotes an operator that returns the weighted average of lagged values of a time-series variable with the probability distribution of incubation periods, $\{p_m\}_{m=1}^{M}$, for the weights:

$$F(Y_t) = \sum_{m=1}^{M} Y_{t-m} p_m, \tag{8}$$

for an arbitrary time-series variable $Y_t$. The second term in Eq (7), $\epsilon_t$, is the residual due to the error term in Eq (5). This term is assumed to be an AR(1) process with a normally-distributed innovation, in order to allow serial correlation of residuals:

$$\epsilon_t = \rho \epsilon_{t-1} + \eta_t, \quad \eta_t \sim N(0, \sigma^2). \tag{9}$$

## Estimation method

The regression model is estimated by the Bayesian method, given the following parameter restrictions and the prior distribution of the initial value of $\epsilon_t$ denoted by $\epsilon_0$:

$$\theta_j < 0, \quad \gamma_j + \theta_j > 0, \quad \gamma_j + \theta_j + \psi_{ij} > 0, \tag{10}$$

$$\rho \in (-1, 1), \tag{11}$$

$$\epsilon_0 \sim N\left(0, \frac{\sigma^2}{1-\rho^2}\right), \tag{12}$$

for $i = 1, 2, \ldots, I$ and $j = 1, 2, \ldots, J$. The prior distribution of the other parameters than $\epsilon_0$ is an improper prior, i.e., a constant for any set of values of the parameters.

The parameter restrictions in Eq (10) are due to an assumption that consumer activities and mobility in public transportation contribute to the spread of COVID-19 in any condition, whereas their infectiousness is decreasing in the degree of absolute humidity. As will be explained in the next section, the assumption on the effect of absolute humidity is based on [38]. The parameter restriction in Eq (11) is the stationarity condition for the AR(1) process of $\epsilon_t$. In Eq (12), the prior distribution of $\epsilon_0$ is assumed to be the unconditional probability distribution of $\epsilon_t$. R ver. 4.1.1 and CmdStan ver. 2.28.1 are used for the estimation of the regression model [39, 40]. The convergence of Markov chain Monte Carlo (MCMC) simulations is confirmed by the function named cmdstan_diagnose() in the CmdStanR package of R. The code and the dataset used in this study are available at https://github.com/hajimetomura/COVID19_household_exp.

## Data

**Dependent variable.**   The maximum length of an incubation period, *M*, is set to 14 days, which is consistent with the guidance published by the Ministry of Health, Labour and Welfare of the Japanese government [41]. Therefore, the dependent variable is the log difference over seven days in the number of new confirmed cases of COVID-19. This feature of the dependent variable is convenient, as it eliminates the day-of-the-week effect on the dependent variable by comparing the numbers of new confirmed cases on the same days of two consecutive weeks. The daily number of new confirmed cases in Japan is published by the Ministry of Health, Labour and Welfare of the Japanese government [42].

**Sample period of the dependent variable for the estimation of the regression model.** For the estimation of the regression model, the sample period of the dependent variable is set from March 1, 2020, to February 1, 2021. The samples before this period are dropped because the number of new confirmed cases was not continuously positive at the beginning of the pandemic.

The end of the sample period is also limited because of the spread of mutant strains in 2021 in Japan. The first cases of mutant strains in the country were confirmed on December 25, 2020 [43]. As of February 19, 2021, 173 cases of mutant strains had been confirmed, including 43 cases found in airport quarantine [44]. Thus, a wide spread of mutant strains was not observed in Japan before the end of January 2021. By using explanatory variables only up to the end of January 2021, this study aims to avoid a possible structural break in the regression due to the spread of mutant strains in 2021. Because of a lag between explanatory variables and the dependent variable in the regression, the end of the sample period of the dependent variable is set to February 1, 2021.

**Classification of household expenditures among explanatory variables.**   For explanatory variables, this study uses average household expenditures per household on each date among households with two or more persons. The data source is the Family Income and Expenditure Survey published by the Ministry of Internal Affairs and Communications of the Japanese government [45]. There is no data for single-person households at daily frequency in this survey. Daily data for each month in this survey are published in the month after next. Thus, household expenditure data are available up to December 31, 2021, as of February 2022.

To construct explanatory variables, household expenditures are classified into eight items: meals at bars and restaurants; soft drinks, confectioneries, and fruits at bars and restaurants; alcoholic drinks at bars and restaurants; non-packaged lodging; domestic travel packages; admissions, viewing, and game fees; clothing and footwear; and the other household

consumption expenditures. Admissions, viewing, and game fees is the sum of admission fees for theaters, museums, stadiums, sports facilities, and theme parks. (See S1 Table for the original Japanese name of each category of household expenditures in the dataset.) The first six items were the subjects of government interventions during the four states of emergency in Japan before December 2021: from April 7, 2020, to May 25, 2020; from January 7, 2021, to March 21, 2021; from April 25, 2021, to June 20, 2021; and from July 12, 2021, to September 30, 2021 [46–49]. More specifically, the government shortened the opening hours of bars and restaurants in populated prefectures, such as Tokyo, in each state of emergency, and prohibited the sales of alcoholic drinks at bars and restaurants entirely in those prefectures during the third and fourth state of emergency. Also, the government closed large-scale indoor facilities or reduced the number of audiences per event during each state of emergency. The admission fees for these facilities and events are included in admissions, viewing, and game fees. On tourism, the government called for self-restraint on traveling during each state of emergency. However, the government also subsidized the expenses of accommodation and transportation in domestic travel packages from July 22, 2020, to December 27, 2020, in order to make up for reduced revenue for the tourism industry. This nationwide subsidy program was called a "Go-To-Travel" campaign [28]. Given these observations, this study includes the first six items listed above in the explanatory variables, in order to estimate associations between consumer activities intervened in by the government and the spread of COVID-19.

Household expenditures on clothing and footwear are separated from the other household consumption expenditures among the explanatory variables, because this category of household expenditures exhibited a high cross correlation with the dependent variable during the sample period (Table 1). The maximum correlation coefficient for clothing and footwear was almost as high as that for alcoholic drinks at bars and restaurants. Because the other large categories of household expenditures did not show as high a cross correlation with the dependent variable as clothing and footwear, household expenditures other than the first seven items listed above are summed into "the other household consumption expenditures" to form one explanatory variable.

**Table 1. Cross-correlation coefficients between the log difference over seven days in the number of new confirmed cases and lagged values of each large category of household expenditures per household in Japan.**

| Large category of household expenditures | Maximum correlation coefficient | Corresponding lag of household expenditures |
|---|---|---|
| Food | 0.17 | 9 |
| Housing | 0.09 | 18 |
| Fuel, light, and water charges | 0.07 | 1 |
| Furniture and household utensils | 0.16 | 10 |
| Clothing and footwear | 0.36 | 9 |
| Medical care | 0.16 | 1 |
| Transportation and communication | 0.15 | 6 |
| Education | 0.25 | 4 |
| Culture and recreation | 0.22 | 9 |
| The other household consumption expenditures | 0.24 | 9 |
| Alcoholic drinks at bars and restaurants | 0.38 | 9 |

Notes: The table shows the maximum cross correlation coefficients between the log difference over seven days in the number of new confirmed cases and lagged values of each large category of nominal household expenditures per household up to 28-day lag in Japan. In the last row, the figures for alcoholic drinks at bars and restaurants are included for comparison, even though it is not part of large categories. The sample period of the log difference over seven days in the number of new confirmed cases is from March 1, 2020, to February 1, 2021.

**Conversion of nominal household expenditures into "real" values in the construction of explanatory variables.** As described above, there was a subsidy program for domestic tourism called a Go-To-Travel campaign from late July to late December 2020 in Japan. While this campaign subsidized accommodation costs for domestic travels, transportation costs were subsidized only if they were packaged with accommodation costs [50]. Therefore, households increased mainly the purchases of domestic travel packages during the campaign period to maximize the amounts of subsidies they received. See S1 Fig to confirm the difference between nominal household expenditures (i.e., the amounts of money paid by households) for non-packaged lodging and domestic travel packages during the campaign period.

To exhibit the subsidy effect of the Go-To-Travel campaign, Fig 1 shows the consumer price indices related to the components of household expenditures included in the explanatory variables. The consumer price index is the average price level for each category of consumer goods and services published by the Ministry of Internal Affairs and Communications of the Japanese government [51]. In the figure, the subsidy effect of the Go-To-Travel campaign is clearly visible in a drop in the consumer price index for lodging during the campaign period, even though this index includes both the prices of non-packaged lodging and the costs of accommodation in domestic travel packages. Thus, households could consume more travel services by buying domestic travel packages with the same amount of money during the campaign period.

For a measure of consumer activities associated with the spread of COVID-19, the amount of packaged travel services consumed by households is more suitable than the nominal value of subsidized consumer spending on domestic travel packages. One way to remove the subsidy effect of the Go-To-Travel campaign to measure the amount of packaged travel services consumed by households is to divide nominal household expenditures on domestic travel packages by the consumer price index of the corresponding category. The resulting value is an artificial measure of quantity, as each nominal expenditure is the product of the quantity and the price of goods or services purchased. In the terminology of economics, this value is called a "real" value. For consistency, real values are used for all the explanatory variables on the classified components of household expenditures per household. See S1 Appendix for more details on how to construct the consumer price index for each explanatory variable. Fig 2 depicts the real values of household expenditures per household used for explanatory variables. For each explanatory variable, the unit of the real value is normalized to 100 yen at the average price in 2020—that is, one unit of real value equals the quantity of consumer goods and services that could be purchased by 100 yen in 2020.

Fig 1 implies that the consumer price indices for the other types of goods and services than lodging did not change much during the sample period. Also, it is described in S1 Appendix that the subsidy effect of the Go-To-Travel campaign on the consumer price index for lodging is removed when the index is used to compute the real values of household expenditures on non-packaged lodging. Thus, the conversion of nominal household expenditures into real values makes a substantial difference only for domestic travel packages.

**Measure of mobility in public transportation.** For the measure of mobility in public transportation among the explanatory variables, this study uses "transit_stations" for Japan in the COVID-19 Community Mobility Reports published by Google [52]. This variable measures a percentage change in mobility in public transportation on each date from the average in the benchmark period from January 3, 2020, to February 6, 2020. Fig 3 depicts the values of this variable.

**Nationwide dummy for absolute humidity.** To construct the dummy variable for absolute humidity in Eq (6), $D_{AH,t}$, a dummy for absolute humidity being no less than $9g/m^3$ is computed for the capital of each prefecture in Japan on each date, using Celsius temperature

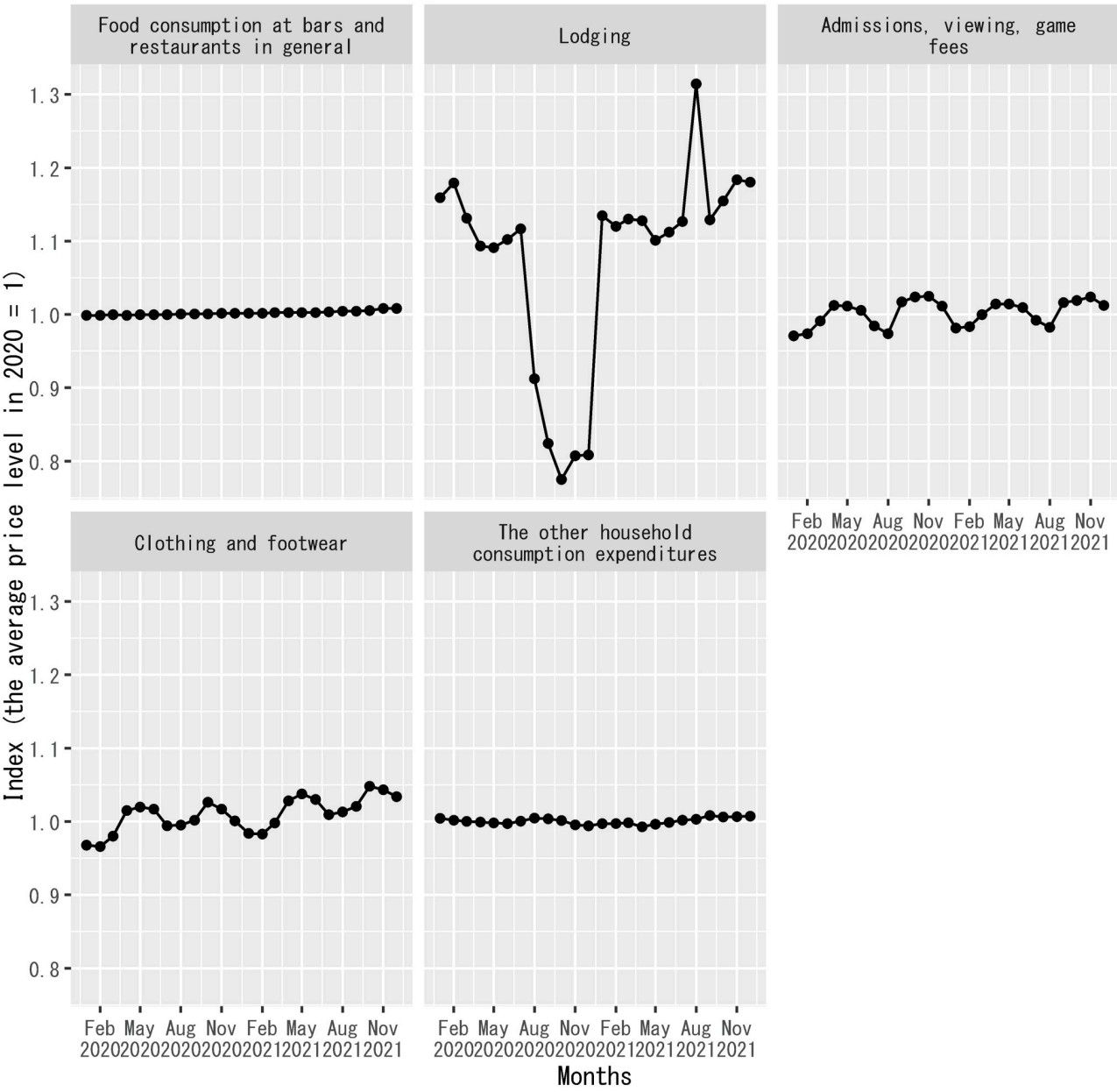

**Fig 1. Price levels of classified components of household expenditures in Japan.** This figure shows the monthly values of the consumer price index related to each component of household expenditures included in the explanatory variables. The sample period shown in the figure is from January 2020 to December 2021. The index for lodging includes both the prices of non-packaged lodging and the costs of accommodation in domestic travel packages. The index for food consumption at bars and restaurants in general covers the prices of three types of food consumption at bars and restaurants included in the explanatory variables. There are no separate consumer price indices corresponding to non-packaged lodging, domestic travel packages, and the three types of food consumption at bars and restaurants in the data source [51].

and relative humidity published by the Japan Meteorological Agency [53]. (See S2 Appendix for the formula to compute absolute humidity, and S3 Appendix for the details about how to fulfill missing values for some prefectures in the dataset.) These dummies are weighted by the population of each prefecture in 2019, and then summed across prefectures to compute the population-weighted nationwide average of the prefectural dummies on each date [54]. Fig 4 depicts the resulting series of the nationwide dummy for absolute humidity.

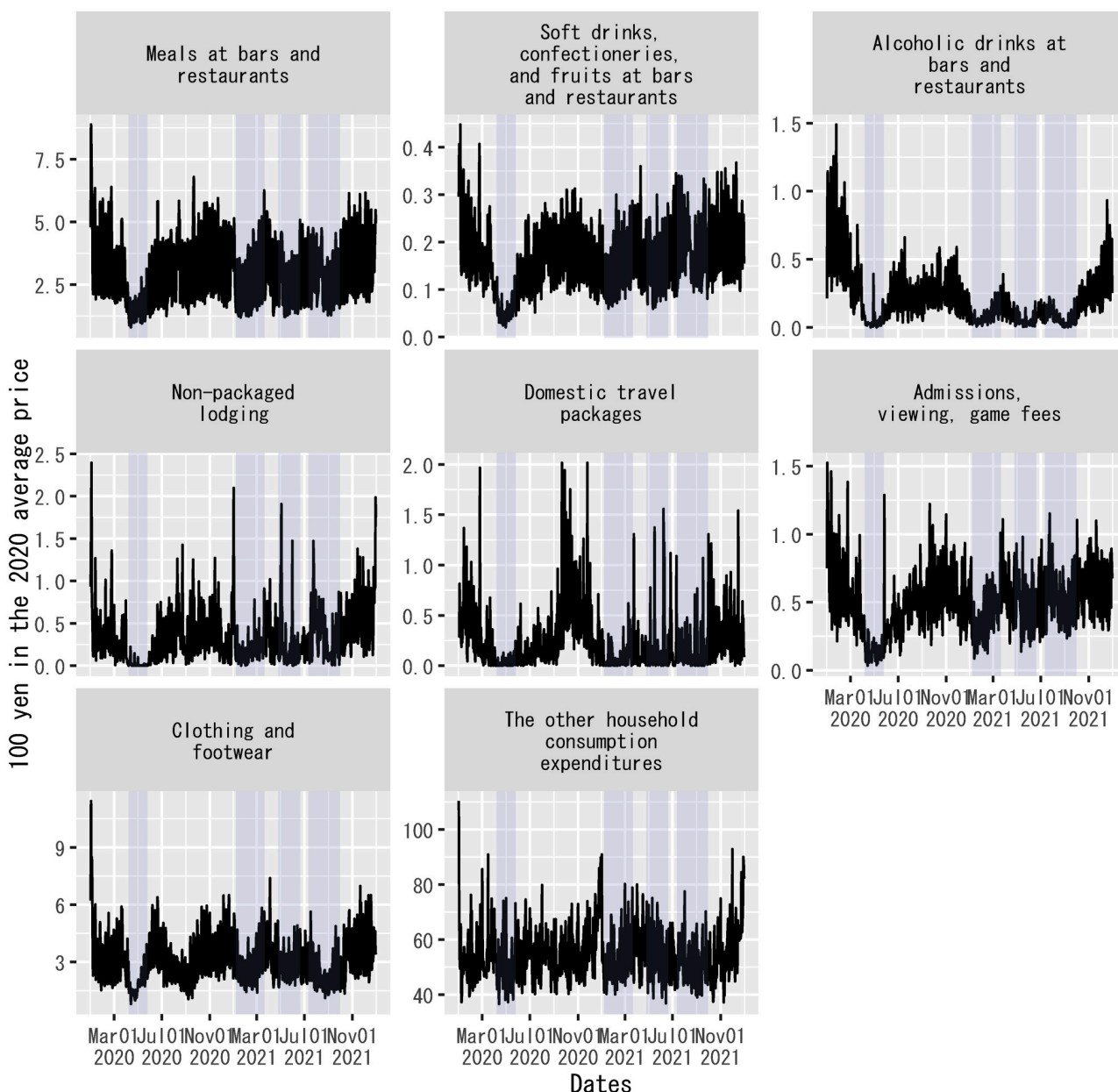

**Fig 2. "Real" values of classified components of household expenditures per household in Japan.** These series are computed by dividing each component of nominal household expenditures per household by the corresponding consumer price index on each date. They are used for explanatory variables in the regression model. The unit of each series is normalized to 100 yen at the average price of each component of household expenditures in 2020. The sample period shown in the figure is from January 1, 2020, to December 31, 2021. Each shadowed period indicates a state of emergency.

The threshold level of absolute humidity for prefectural dummies, $9g/m^3$, is adopted from a study that found that the risk ratio of new infections of COVID-19 had a non-linear relationship with absolute humidity in England, peaking around $6–8g/m^3$ [38]. Prefectural dummies for absolute humidity approximate this non-linear relationship at the capital of each prefecture by a step function.

**Time dummies.** In Eq 6, the time dummy for the year-end and new-year holiday period, $D_{NY,t}$, is set to unity for the period from December 29, 2020, to January 3, 2021. Time dummies

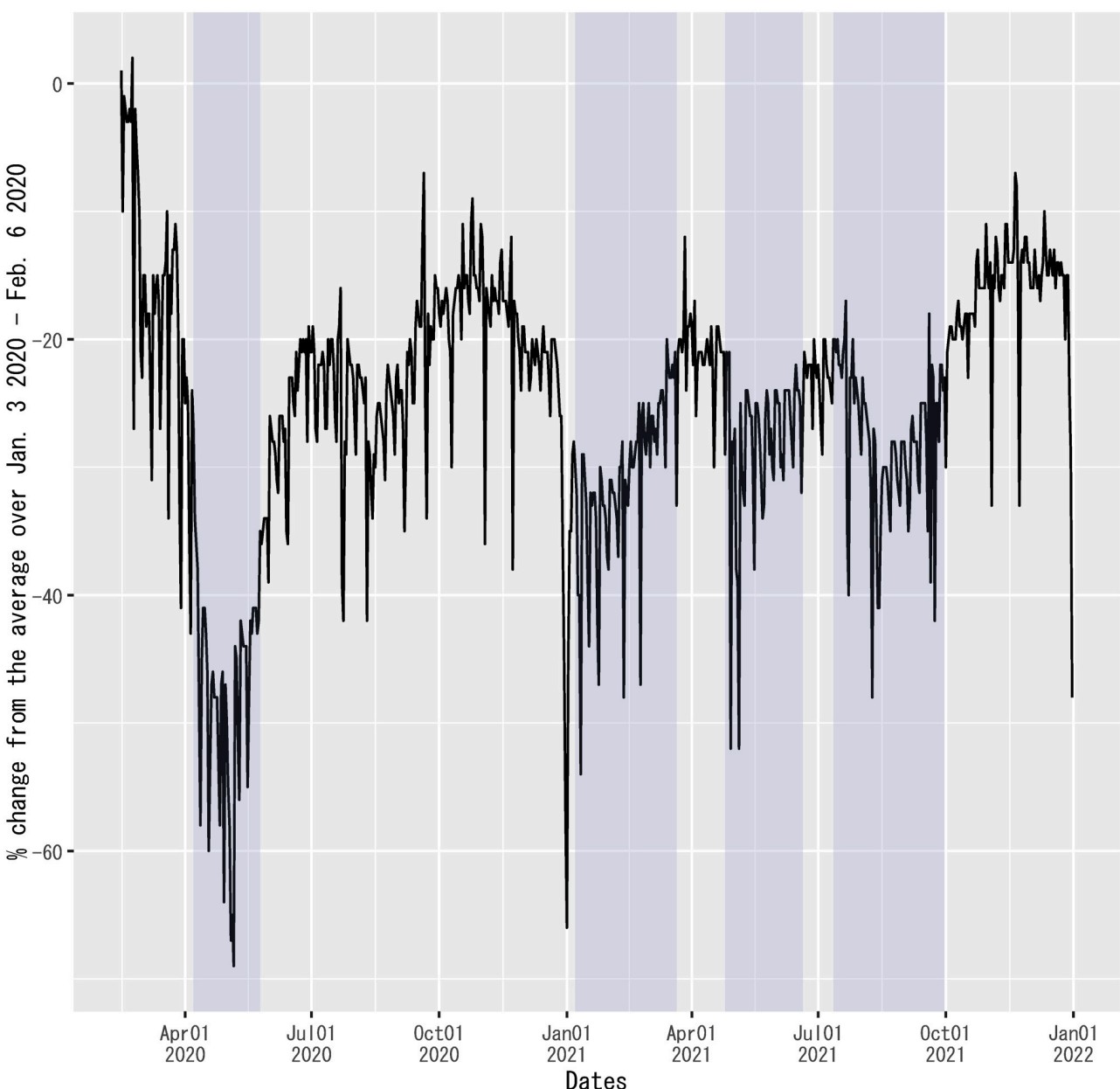

**Fig 3. Mobility in public transportation in Japan.** The figure shows transit_stations for Japan in the COVID-19 Community Mobility Reports [52]. The sample period shown in the figure is from February 15, 2020, to December 31, 2021. Each shadowed period indicates a state of emergency.

related to states of emergency, $D_{SoE,i,t}$ for $i = 1, 2, 3$, are set to unity for each of the following periods separately: the period from the beginning of the sample period to April 6, 2020; the first state of emergency from April 7, 2020, to May 25, 2020; and the second state of emergency from January 7, 2021, to March 21, 2021. The time dummy for the second state of emergency is defined up to the end of the sample period of explanatory variables for the estimation of the regression model, i.e., January 31, 2021.

**Sample distribution of incubation periods.** For the probability distribution of incubation periods in Eq (8), $\{p_m\}_{m=1}^{14}$, I use the sample distribution of incubation periods in Japan reported by [6]. Fig 5 depicts the sample distribution.

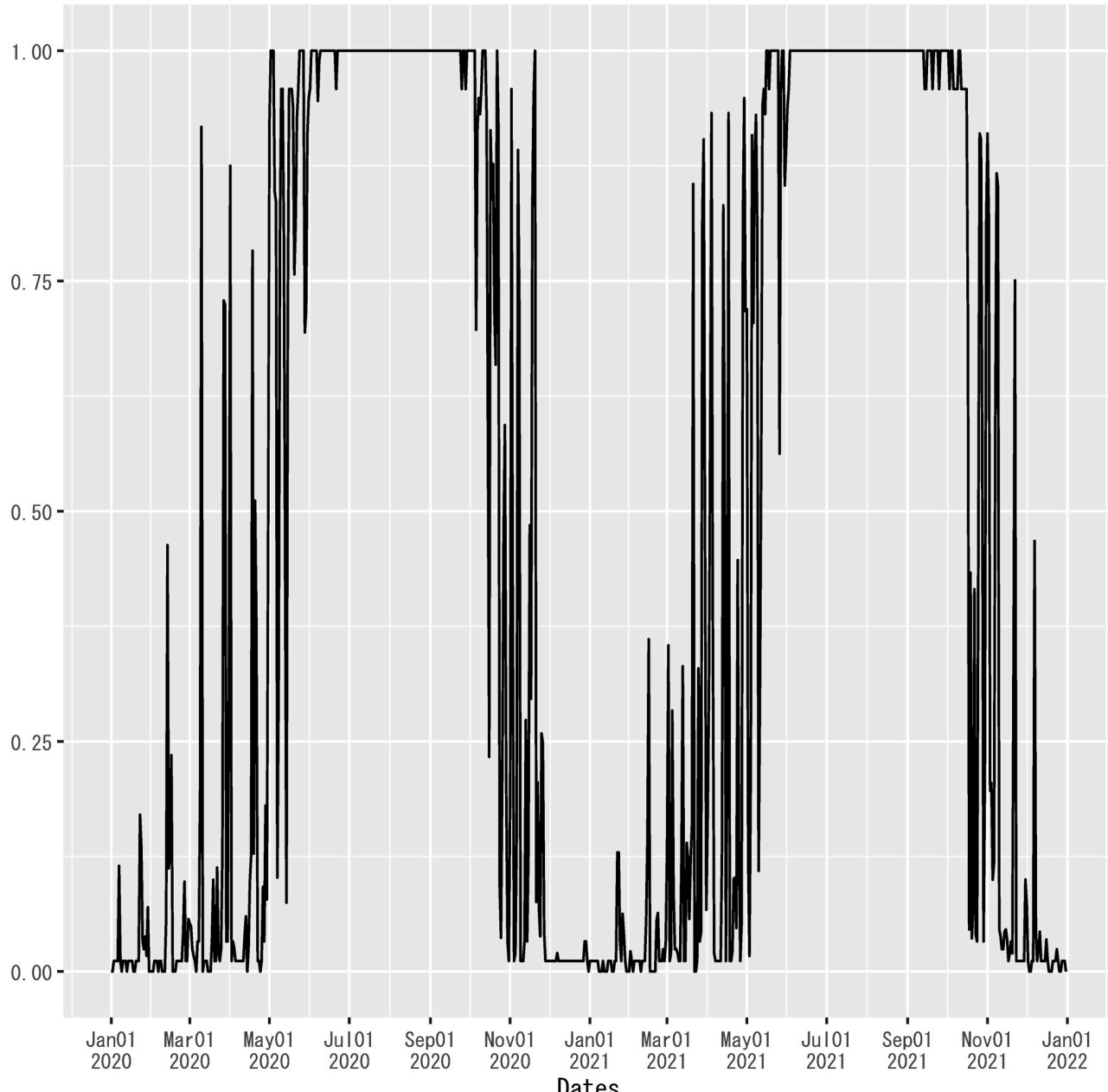

**Fig 4. Nationwide dummy for absolute humidity in Japan.** This series is the population-weighted nationwide average of prefectural dummies for absolute humidity being no less than $9 g/m^3$ at the capital of each prefecture on each date. The sample period shown in the figure is from January 1, 2020, to December 31, 2021.

## Sensitivity analysis

Associations between weather conditions and COVID-19 incidence and transmission have been examined across countries [55]. For example, it has been reported in a global study that approximately 85% of the COVID-19 cases reported before May 1, 2020, occurred in regions with outside temperature between 3 and 17˚C and absolute humidity between 1 and $9 g/m^3$, the latter of which is consistent with the construction of the nationwide dummy for absolute

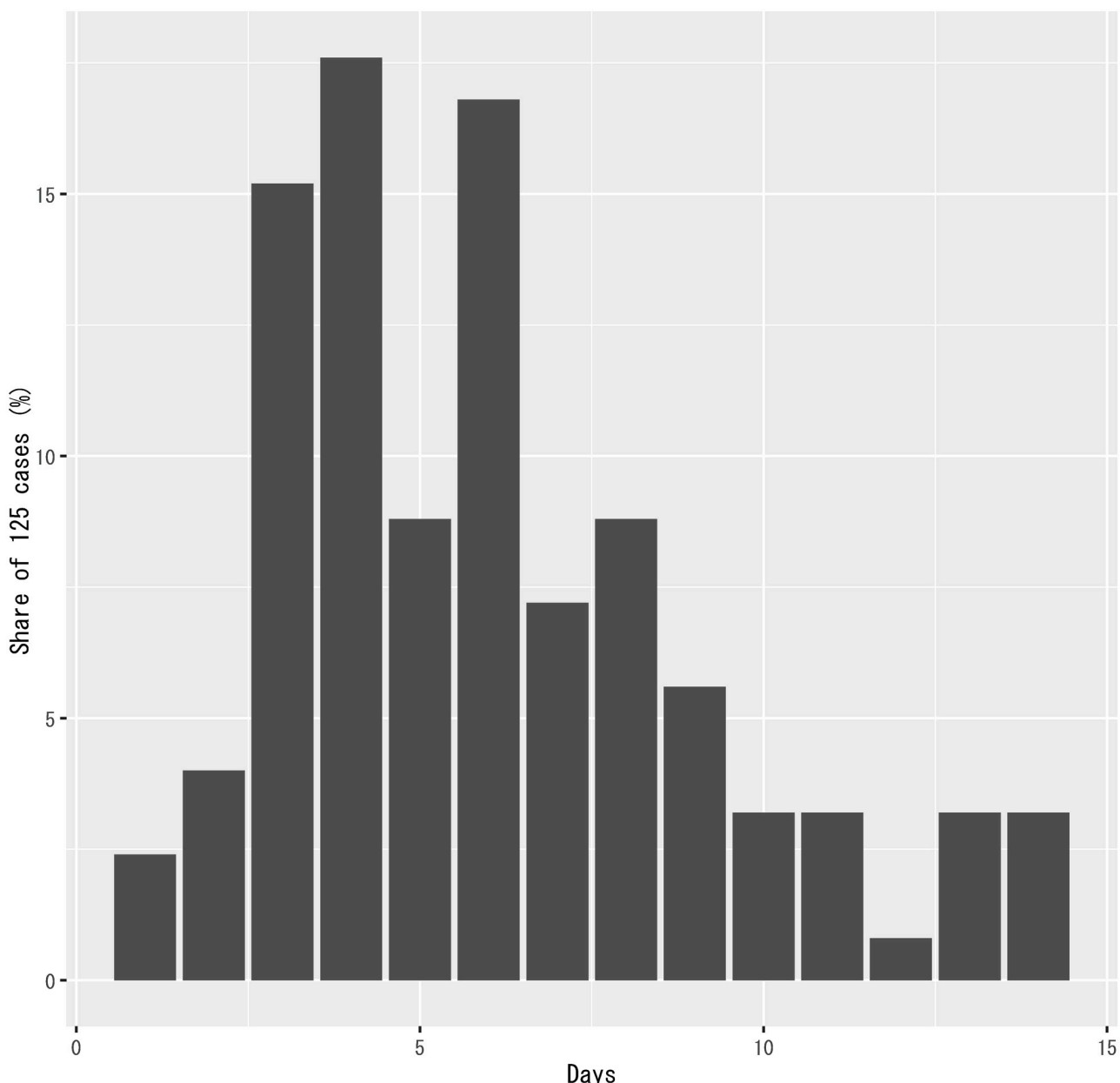

**Fig 5. Sample distribution of incubation periods in Japan reported by [6].**

humidity described above [56]. For sensitivity analysis, the regression model is estimated with not only the baseline set of explanatory variables described above, but also four alternative sets in each of which the nationwide dummy for absolute humidity is removed from the set of explanatory variables, or substituted by one of the following alternative weather variables: a nationwide dummy for outside temperature being no less than $18°C$; the nationwide average of outside temperature; and the nationwide average of absolute humidity. All of the nationwide dummy and averages are population-weighted across prefectures in Japan in the same way as the construction of the nationwide dummy for absolute humidity. See S4 Appendix for more details on the construction of alternative weather variables.

In addition, among the classified components of household expenditures included in the explanatory variables, household expenditures on clothing and footwear are purchases of merchandise goods. The other components of household expenditures are purchases of offline services, except "the other household consumption expenditures", which represents the residual household expenditures. Because the infectiousness of consumer activities can be different between online and offline merchandise shopping, the regression model is estimated with an alternative set of explanatory variables in which household expenditures on clothing and footwear only include offline purchases on each date. This variable is computed by multiplying the real value of household expenditures on clothing and footwear per household on each date by the monthly offline share of the expenditures in the same month. The monthly offline share can be computed from data available from the Survey of Household Economy published by the Ministry of Internal Affairs and Communications of the Japanese government [57]. See S5 Appendix for more details on the construction of offline household expenditures on clothing and footwear.

## Results

### Estimated regression coefficients

Table 2 shows the estimated regression coefficients of classified components of household expenditures and mobility in public transportation, except the coefficients of cross terms with time dummies related to states of emergency. See S2 Table for the estimation results for all the parameters in the regression model.

The linear coefficient of each explanatory variable (i.e., "Baseline effect" in Table 2) estimates the degree of an association between the explanatory variable and the rate of change in the number of new confirmed cases of COVID-19 when absolute humidity is less than $9g/m^3$ at the capitals of all prefectures (i.e., the nationwide dummy for absolute humidity is zero), whereas the cross effect of absolute humidity measures how much the degree of this association changes if absolute humidity becomes equal to, or greater than, $9g/m^3$ at the capitals of all prefectures (i.e., the nationwide dummy for absolute humidity becomes one). For each component of household expenditures, the regression coefficient is measured per the real value of

**Table 2. Estimated regression coefficients of classified components of household expenditures and mobility in public transportation in Japan.**

| Explanatory variable | Baseline effect | | | Cross effect of absolute humidity | | |
|---|---|---|---|---|---|---|
| | Posterior mean | 2.5th percentile | 97.5th percentile | Posterior mean | 2.5th percentile | 97.5th percentile |
| Meals at bars and restaurants | 0.060 | 0.006 | 0.167 | -0.018 | -0.068 | -0.000 |
| Soft drinks, confectioneries, and fruits at bars and restaurants | 1.038 | 0.126 | 2.914 | -0.394 | -1.432 | -0.009 |
| Alcoholic drinks at bars and restaurants | 1.339 | 0.172 | 3.120 | -0.231 | -0.844 | -0.006 |
| Non-packaged lodging | 0.356 | 0.047 | 0.912 | -0.135 | -0.481 | -0.003 |
| Domestic travel packages | 0.221 | 0.029 | 0.608 | -0.100 | -0.361 | -0.002 |
| Admissions, viewing, and game fees | 0.102 | 0.012 | 0.261 | -0.022 | -0.078 | -0.001 |
| Clothing and footwear | 0.266 | 0.029 | 0.768 | -0.120 | -0.468 | -0.003 |
| The other household consumption expenditures | 0.006 | 0.001 | 0.017 | -0.002 | -0.005 | -0.000 |
| Mobility in public transportation | 0.019 | 0.004 | 0.042 | -0.012 | -0.029 | -0.001 |

Notes: The dependent variable is the log difference over seven days in the number of new confirmed cases of COVID-19 in Japan. The sample period of the dependent variable is from March 1, 2020, to February 1, 2021. Each row corresponds to the explanatory variable in the first column. In the rest of columns, "Baseline effect" indicates the linear coefficient of the explanatory variable, $\gamma_j$, whereas "Cross effect of absolute humidity" indicates the coefficient of the cross term between the explanatory variable and the nationwide dummy for absolute humidity, $\theta_j$, in Eq (6).

spending per household. Thus, the posterior means of regression coefficients in the table indicate that alcoholic drinks at bars and restaurants had the highest association with the spread of COVID-19 per the real value of spending per household, regardless of the degree of absolute humidity. Soft drinks, confectioneries, and fruits at bars and restaurants had the second highest association.

## Out-of-sample forecast performance of the regression

Fig 6 plots the observed values of the dependent variable, the fitted values of the regression up to February 1, 2021, and the out-of-sample forecasts of the regression from February 2, 2021, onward. The out-of-sample forecasts are computed by inserting into the regression the values of explanatory variables during the out-of-sample forecast period, given each MCMC sample of parameter values, except the time dummy for the second state of emergency. For the out-of-sample forecast period, the time dummy for the second state of emergency is set to zero, because otherwise the out-of-sample forecasts would overpredict the observed values of the dependent variable significantly (see S2 Fig). As will be described below, this is likely due to the overfitting of regression coefficients for the time dummies related to states of emergency. Given this set-up, the figure exhibits that the observed values of the dependent variable traced the out-of-sample forecasts by the posterior means of regression coefficients closely until mid-June 2021. $R^2$ for the out-of-sample forecasts from February 2, 2021, to June 30, 2021, is 0.60.

On the other hand, the fitted values continuously deviate from the observed values of the dependent variable for the summer of 2020, when there was a surge in the number of new confirmed cases, so-called the "second wave" in Japan, following the first surge in the spring of 2020.

## Decomposition of out-of-sample forecasts of the regression

Given the good fit of out-of-sample forecasts by the posterior means of regression coefficients up to mid-June 2021, Fig 7 shows the contribution of each explanatory variable to the dependent variable in the regression, which is measured by the product of the explanatory variable and the posterior mean of the corresponding regression coefficient during the out-of-sample forecast period.

Fig 7 plots each explanatory variable's contribution to the dependent variable without the cross effect of absolute humidity, as well as the total contribution including the cross effect of absolute humidity. The figure demonstrates that a higher degree of absolute humidity reduced the absolute size of each explanatory variable's contribution to the dependent variable, as it weakened the association between each explanatory variable and the spread of COVID-19. At the same time, the figure also indicates that the main cause of fluctuations in each explanatory variable's contribution to the dependent variable was not changes in absolute humidity, but idiosyncratic fluctuations in the explanatory variable. This observation presents evidence against reverse causality such that the dependent variable caused concerted fluctuations in the explanatory variables as a confounder, resulting in a good fit of out-of-sample forecasts of the regression.

Fig 8 plots each explanatory variable's contribution to the dependent variable on each date in the form of the difference from the beginning of the out-of-sample forecast period. The figure demonstrates the contribution of each explanatory variable to changes in the dependent variable. For the period before mid-June 2021, the figure indicates that the increase in the

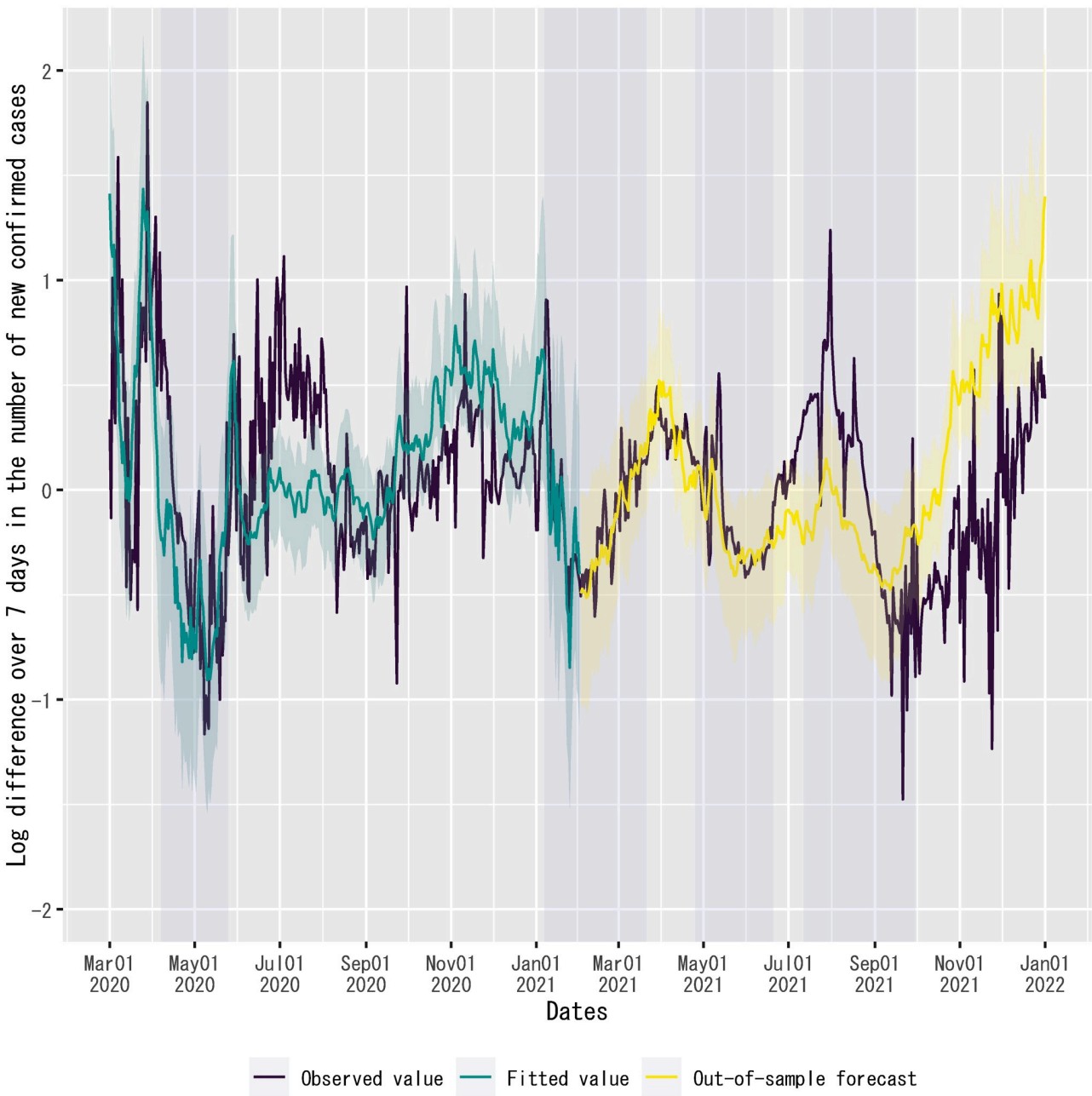

**Fig 6. Fitted values and out-of-sample forecasts of the regression.** The dependent variable is the log difference over seven days in the number of new confirmed cases of COVID-19 in Japan. For out-of-sample forecasts, the time dummy for the second state of emergency is set to zero without changing the posterior means of regression coefficients. The sample period shown in the figure is from March 1, 2020, to January 1, 2022. For the fitted values and the out-of-sample forecasts, the solid line is the posterior mean and the shadowed area indicates the 95% credible interval on each date. Each shadowed period indicates a state of emergency.

number of new confirmed cases around the end of March 2021, which is often called the "fourth wave" in Japan following the three surges in the number of new confirmed cases since the onset of the pandemic, was mainly due to increases in alcoholic drinks at bars and restaurants, non-packaged lodging, clothing and footwear, and mobility in public transportation.

## Decomposition of fitted values of the regression

If the fitted values of the regression are decomposed in the same way as in Fig 7, it can be shown that the cross effects of time dummies related to states of emergency are so large that the other components of contributions of explanatory variables to the dependent variable are almost invisible. Yet, the fitted values of the regression do not change much even if all the time

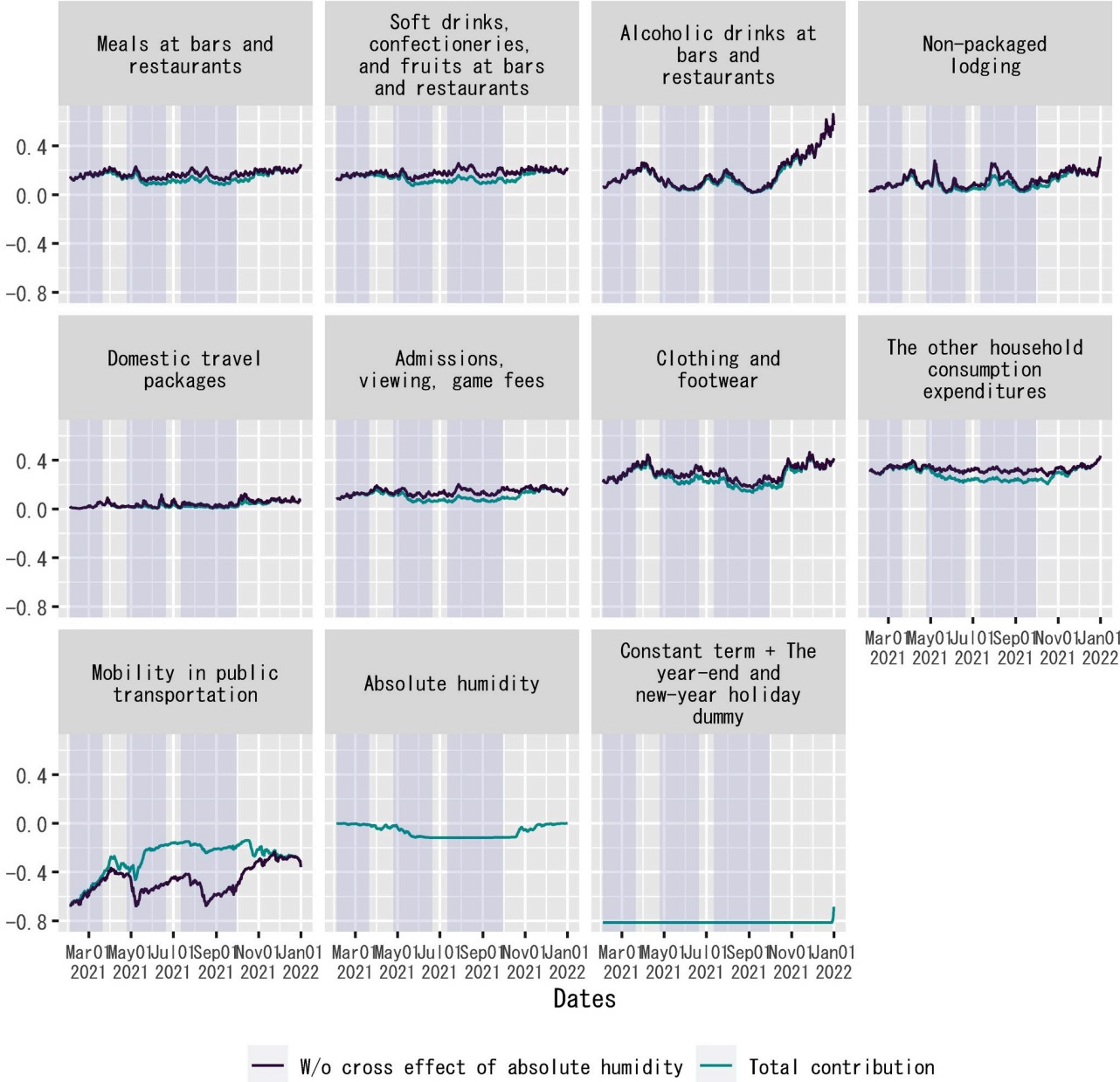

**Fig 7. Decomposition of out-of-sample forecasts of the regression: Level.** Each panel shows the product of an explanatory variable and the posterior mean of the corresponding regression coefficient. For out-of-sample forecasts, the time dummy for the second state of emergency is set to zero without changing the posterior means of regression coefficients. The sample period shown in the figure is from February 2, 2021, to January 1, 2022. For household expenditures and mobility in public transportation, "W/o cross effect of absolute humidity" indicates the posterior mean of $\gamma_j F(X_{j,t})$ in Eq (7), whereas "Total contribution" indicates the posterior mean of $\gamma_j F(X_{j,t}) + \theta_j F(D_{AH,t}X_{j,t})$ in Eq (7) on each date. Each shadowed period indicates a state of emergency.

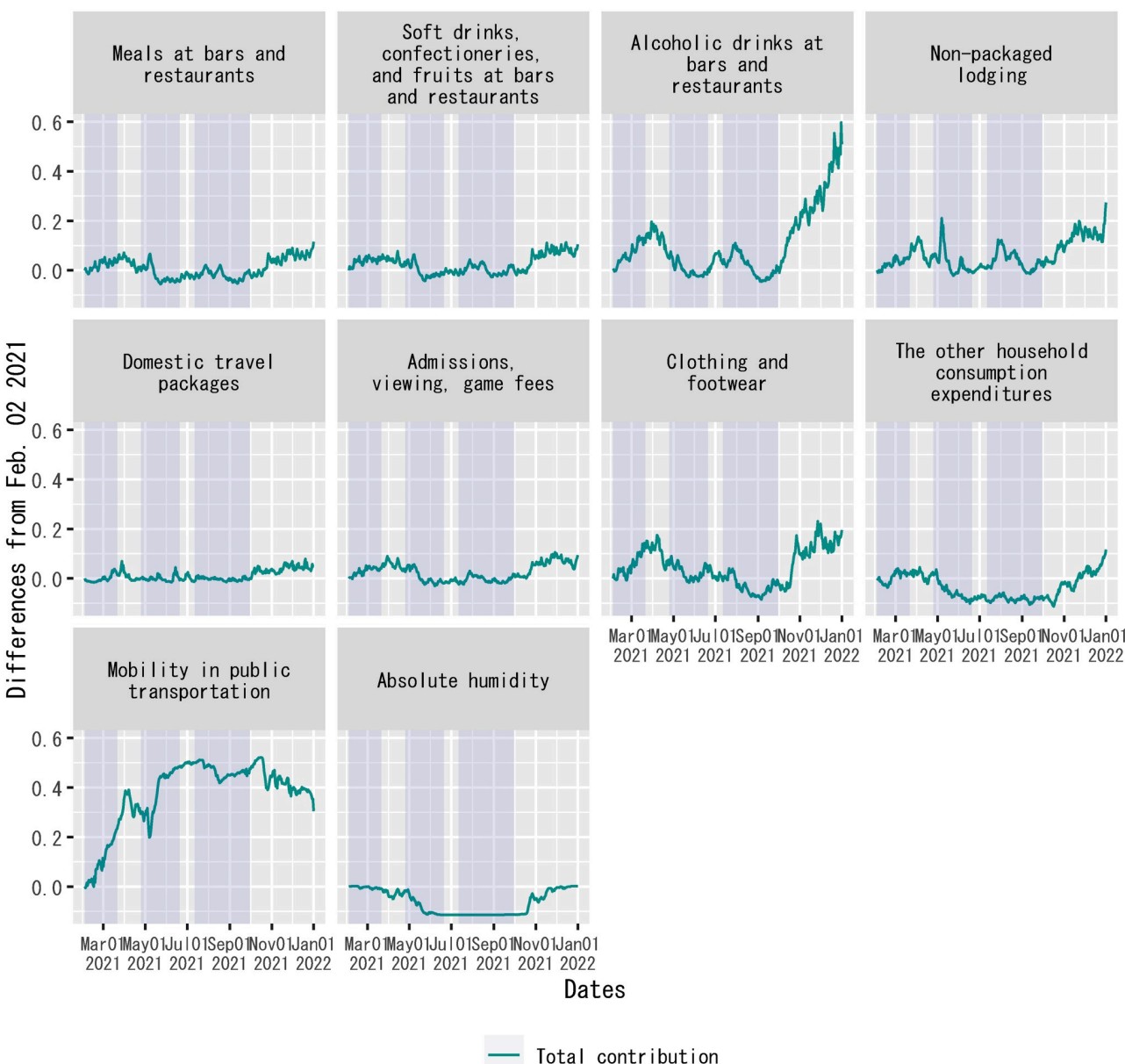

**Fig 8. Decomposition of out-of-sample forecasts of the regression: Differences from a benchmark date.** Each panel plots the differences in the product of an explanatory variable and the posterior mean of the corresponding regression coefficient from February 2, 2021. For out-of-sample forecasts, the time dummy for the second state of emergency is set to zero without changing the posterior means of regression coefficients. The sample period shown in the figure is from February 2, 2021, to January 1, 2022. For household expenditures and mobility in public transportation, "Total contribution" indicates differences in the posterior mean of $\gamma_j F(X_{j,t}) + \theta_j F(D_{AH,t} X_{j,t})$ in Eq (7) on each date from February 2, 2021. Each shadowed period indicates a state of emergency.

dummies related to states of emergency are set to zero, given the same values of posterior means of regression coefficients being used. Thus, the large cross effects of time dummies related to states of emergency offset each other, which is likely due to overfitting. See S3 and S4 Figs for this result.

Given this observation, Fig 9 shows the decomposition of fitted values of the regression when the time dummies related to states of emergency are set to zero without changing the

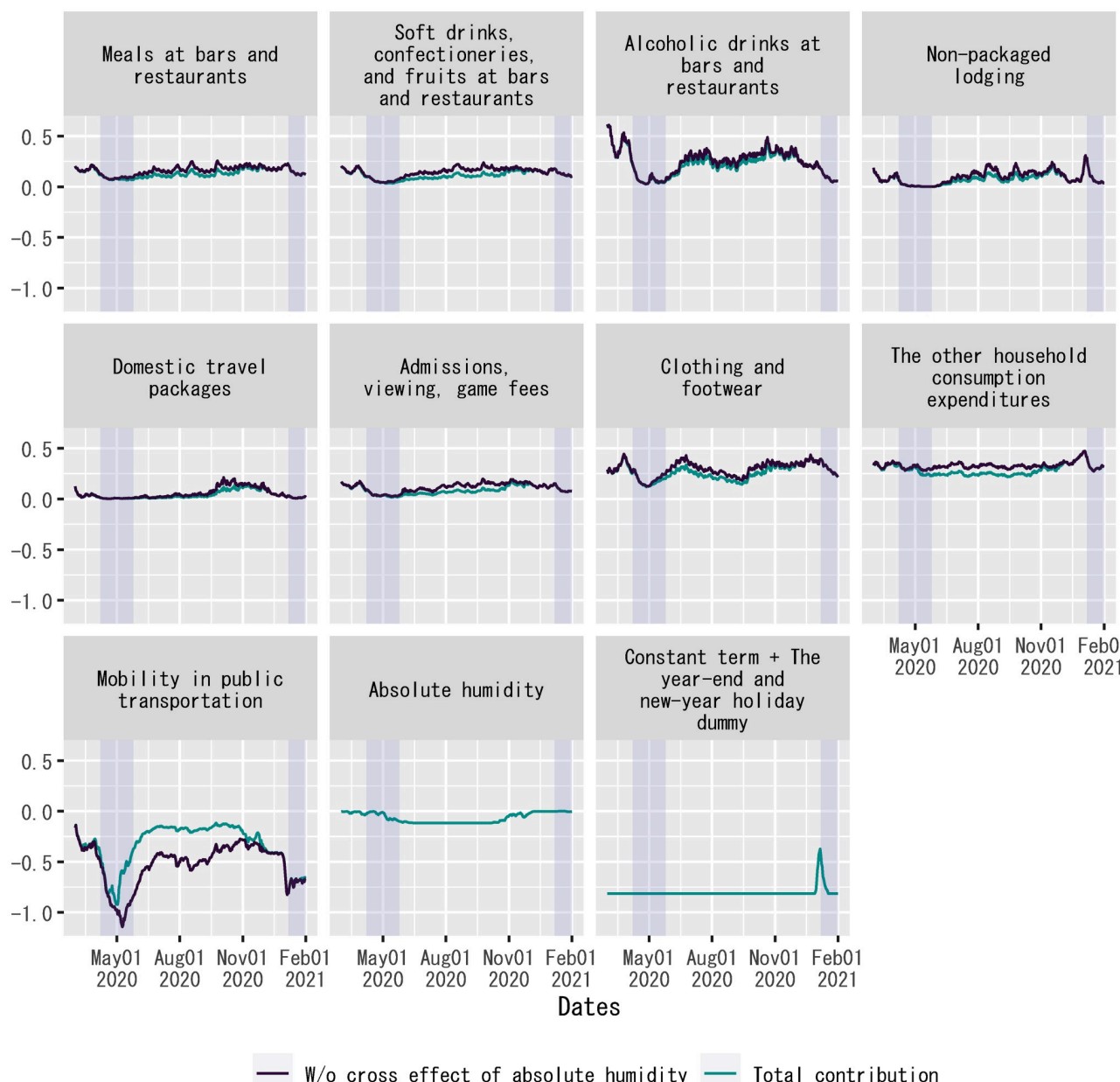

**Fig 9. Decomposition of fitted values of the regression: Level.** Each panel shows the product of an explanatory variable and the posterior mean of the corresponding regression coefficient, when time dummies for the period before the first state of emergency, the first state of emergency, and the second state of emergency are set to zero without changing the posterior means of regression coefficients. The sample period shown in the figure is from March 1, 2020, to February 1, 2021. For household expenditures and mobility in public transportation, "W/o cross effect of absolute humidity" indicates the posterior mean of $\gamma_j F(X_{j,t})$ in Eq (7), whereas "Total contribution" indicates the posterior mean of $\gamma_j F(X_{j,t}) + \theta_j F(D_{AH,t} X_{j,t})$ in Eq (7) on each date. Each shadowed period indicates a state of emergency.

posterior means of regression coefficients. For this case, Fig 10 also plots each explanatory variable's contribution to the dependent variable in the form of the differences from February 1, 2021, i.e., the end of the sample period for the estimation of the regression model. Fig 10 indicates that fluctuations in alcoholic drinks at bars and restaurants, clothing and footwear, and

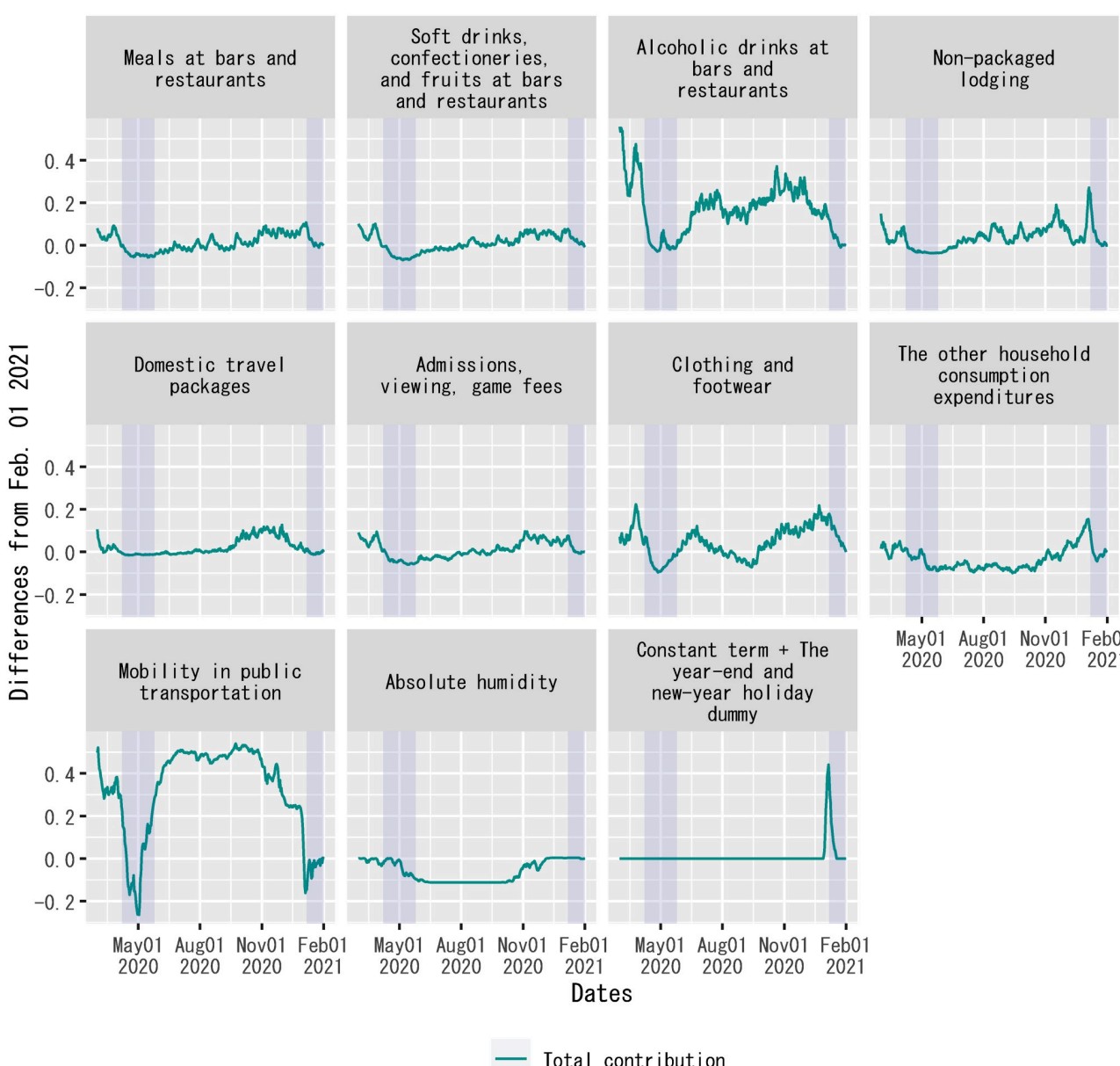

**Fig 10. Decomposition of fitted values of the regression: Differences from a benchmark date.** Each panel shows the differences in the product of an explanatory variable and the posterior mean of the corresponding regression coefficient from February 1, 2021, when time dummies for the period before the first state of emergency, the first state of emergency, and the second state of emergency are set to zero without changing the posterior means of regression coefficients. The sample period shown in the figure is from March 1, 2020, to February 1, 2021. For household expenditures and mobility in public transportation, "Total contribution" indicates differences in the posterior mean of $\gamma_j F(X_{j,t}) + \theta_j F(D_{AH,t} X_{j,t})$ in Eq (7) on each date from February 1, 2021. Each shadowed period indicates a state of emergency.

mobility in public transportation made outstandingcontributions to changes in the dependent variable before February 2021.

## Sensitivity analysis

For sensitivity analysis on the effect of weather conditions on COVID-19 incidence and transmission, the regression model is estimated with four alternative sets of explanatory variables in

**Table 3. $R^2$ for the out-of-sample forecasts of regressions with alternative weather variables for explanatory variables.**

| | Type of weather variable included in the explanatory variables | | | | |
| --- | --- | --- | --- | --- | --- |
| | Nationwide dummy for absolute humidity | Nationwide dummy for outside temperature | No weather variable | Nationwide average of absolute humidity | Nationwide average of outside temperature |
| $R^2$ for the out-of-sample forecasts of the regression from February 2, 2021, to June 30, 2021 | 0.60 | 0.48 | 0.57 | 0.31 | 0.43 |

Notes: The first row lists the type of variable included as an explanatory variable in the regression model to represent the effect of weather conditions on the dependent variable. "No weather variable" in the fourth column means that no weather variable is included among the explanatory variables. The second row shows the value of $R^2$ for the out-of-sample forecasts of each regression by the posterior means of regression coefficients from February 2, 2021, to June 30, 2021. See S4 Appendix for more details.

each of which the nationwide dummy for absolute humidity is removed from the set of explanatory variables, or substituted by one of the following weather variables: a nationwide dummy for outside temperature being no less than 18˚C; the nationwide average of outside temperature; and the nationwide average of absolute humidity. See S4 Appendix for the details of the estimation results.

Table 3 shows the value of $R^2$ for the out-of-sample forecasts of each regression by the posterior means of regression coefficients from February 2, 2021, to June 30, 2021. The table implies that each regression shows a good fit of out-of-sample forecasts, including the one without any weather variable among the explanatory variables. Even though the benchmark regression with the nationwide dummy for absolute humidity shows the highest value of $R^2$ among the five regressions listed in the table, there is only a slim difference between the benchmark regression and the regression without a weather variable. Thus, using out-of-sample forecast performance measured by $R^2$ for model evaluation, this study cannot be conclusive on the importance of weather conditions for COVID-19 incidence and transmission.

It is also shown in S4 Appendix that in each regression listed in Table 3, alcoholic drinks at bars and restaurants had the highest association with the dependent variable of the regression per the real value of spending per household among the classified components of household expenditures included in the explanatory variables, whereas soft drinks, confectioneries, and fruits at bars and restaurants had the second highest association. Also, the results of the regressions with the three highest values of $R^2$ shown in Table 3, i.e., those with nationwide dummies for absolute humidity and outside temperature, and no weather variable among the explanatory variables, indicate that the assumption on a weather variable in the explanatory variables mostly affects the estimates of the regression coefficients of mobility in public transportation, leaving unchanged the implication of the regression model for the contribution of each classified component of household expenditures to the spread of COVID-19.

In addition, the regression model is also estimated with another alternative set of explanatory variables in which household expenditures on clothing and footwear only include offline purchases. See S5 Appendix for the details of the estimation result. The value of $R^2$ for the out-of-sample forecasts of the regression by the posterior means of regression coefficients from February 2, 2021, to June 30, 2021 is 0.59, which is almost the same as the value for the benchmark regression described above, 0.60. Also, the decomposition of fitted values and out-of-sample forecasts of the regression is similar to that of the benchmark regression reported in Figs 7–10.

A possible reason for this result is the co-movement of online and offline household expenditures on clothing and footwear during the sample period. As shown in S5 Appendix, the

monthly online share of household expenditures on clothing and footwear had been stable after a permanent rise in early 2020. If this observation held at daily frequency as well, then the average effect of total (i.e, the sum of online and offline) household expenditures on clothing and footwear would be proportional to the effect of the offline household expenditures on each date. This interpretation is consistent with the good fit of out-of-sample forecasts of the benchmark regression reported above, because if the ratio between online and offline household expenditures on clothing and footwear changed significantly on each date, then it would lower the fit of out-of-sample forecasts of the benchmark regression by making the true average effect of total household expenditures on clothing and footwear significantly time-varying. Nonetheless, obtaining daily data on the online share of household expenditures on clothing and footwear is necessary to confirm the legitimacy of this interpretation. This issue remains for future research.

### Preliminary regression of out-of-sample forecast errors on the Delta-variant share of new confirmed cases and the twice-vaccinated share of the population

As shown in Fig 6, the observed values of the dependent variable overshot the out-of-sample forecasts of the regression from mid-June to August 2021. Then, the out-of-sample forecasts overpredicted the dependent variable from September 2021 onward. Even though the regression model does not have any information about the determinants of out-of-sample forecast errors, the beginning of the overshooting coincided with the spread of the Delta variant in Japan since June 2021 (see Fig 11). Because there was also a steady progress of vaccinations in Japan from June 2021, vaccinations might gradually mitigate the effect of the Delta variant, reducing the magnitude of overshooting in August 2021 (see Fig 12).

For preliminary analysis, the out-of-sample forecast errors of the regression on each date is regressed on the Delta-variant share of new confirmed cases and the twice-vaccinated share of the population seven days ago. The lag length of the explanatory variables in this regression is chosen because it corresponds to the mid-point of the range of incubation periods [41]. The Delta-variant share of new confirmed cases in Tokyo is included in the explanatory variables as a proxy for the nationwide share in this regression, because it is available for a longer sample period than the reported nationwide share, as shown in Fig 11. In this regression, the weekly value of this variable is used on each date within the same week, because no daily data are available. The sample period for the estimation of this regression is terminated at the end of October 2021, because the November data for the Delta-variant share of new confirmed cases in Tokyo is based on only five cases due to a decline in the number of new confirmed cases in that month. The estimation method is ordinary least squares.

Table 4 shows the estimation result. There is no constant term among the explanatory variables in the regression, so that the fitted value of the regression equals zero (i.e., the regression does not explain any component of out-of-sample forecast errors) if the two explanatory variables equal zero. A relatively high value of $R^2$ shown in Table 4 implies that the out-of-sample forecast errors can be fitted to a linear combination of the Delta-variant share of new confirmed cases and the twice-vaccinated share of the population. Fig 13 plots the fitted values of the regression of out-of-sample forecast errors, along with the values of out-of-sample forecast errors. The figure implies that, if the out-of-sample forecast errors were associated with the spread of the Delta variant and the progress of vaccinations, then there was an exception from mid-June to July 2021. A caveat is that this implication of the regression is tentative, as the estimation result shown in Table 4 is just an in-sample fit, yet to be validated by out-of-sample forecast performance.

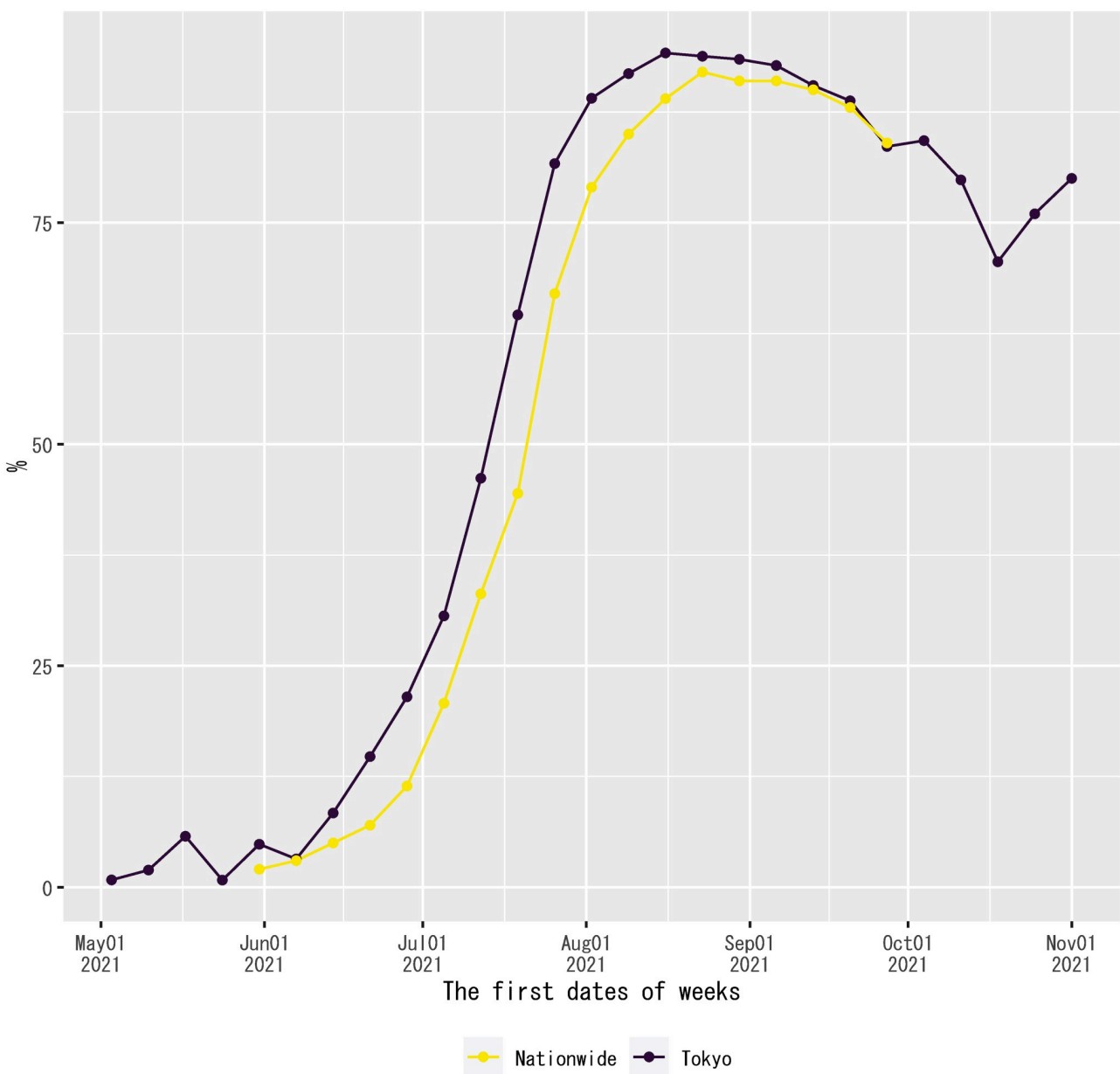

**Fig 11. Delta-variant share of new confirmed cases of COVID-19 in Japan.** Source: The Ministry of Health, Labour and Welfare, the Government of Japan; and the Tokyo Metropolitan Government [58, 59]. Notes: The Ministry of Health, Labour and Welfare and the Tokyo Metropolitan Government stopped reporting the data after the weeks of Sept. 27, 2021, and Nov. 1, 2021, respectively.

## Discussion

### Principal findings

By estimating a time-series regression model, this study has found that there had been stable associations between classified components of household expenditures and the log difference over seven days in the number of new confirmed cases of COVID-19 in Japan before June 2021, except for a low in-sample fit of the regression during the summer of 2020. These associations can be validated using out-of-sample regression forecasts for February and June 2021.

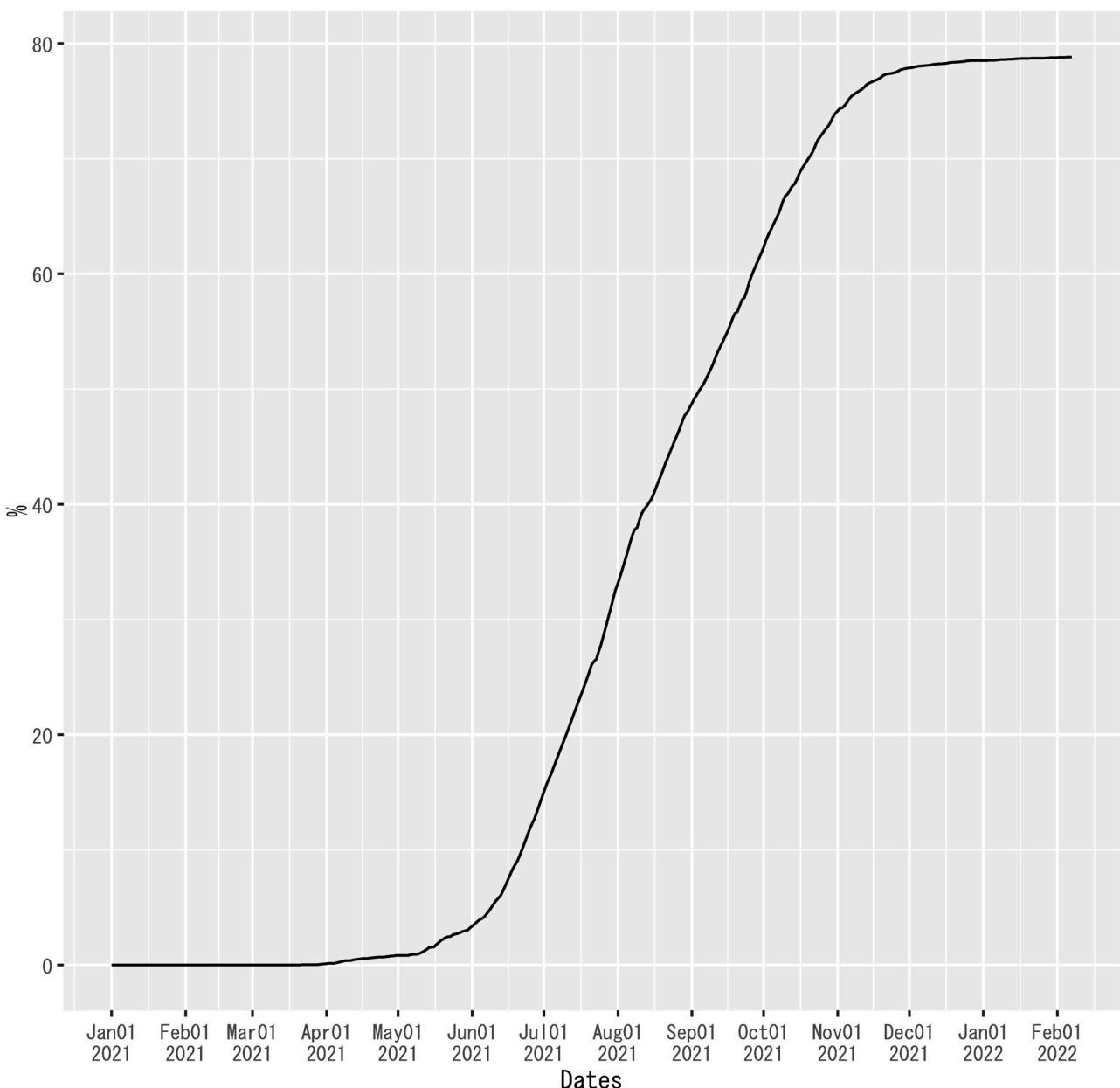

**Fig 12. Twice-vaccinated share of the population in Japan.** Source: The Ministry of Health, Labour and Welfare, the Government of Japan [60].

The estimated regression coefficients indicate that alcoholic drinks at bars and restaurants had the highest association with the rate of change in the number of new confirmed cases per the real value of spending per household among the classified components of household expenditures in the explanatory variables. The second-highest association was soft drinks, confectioneries, and fruits at bars and restaurants. The decomposition of fitted values and out-of-sample forecasts of the regression implies that household expenditures on alcoholic drinks at bars and restaurants, clothing and footwear, and mobility in public transportation made substantial contributions to fluctuations in the rate of change in the number of new confirmed cases in Japan between March 2020, that is, the beginning of the sample period, and June 2021.

**Table 4. Regression of out-of-sample forecast errors of the regression model reported in Table 2.**

| Explanatory variable | Coefficient estimate | Standard error |
|---:|:---:|---:|
| Delta-variant share of new confirmed cases (%, 7 days lag) | 0.014 | 0.001 |
| Twice-vaccinated share of the population (%, 7 days lag) | -0.026 | 0.001 |

Number of observations: 272. $R^2$: 0.44

Notes: The dependent variable is the out-of-sample forecast errors generated by the regression model with the posterior means of regression coefficients reported in S2 Table. The lag of each explanatory variable is seven days. There is no constant term in the regression. The sample period for the dependent variable is from February 2, 2021, to October 31, 2021. The estimation method is ordinary least squares.

The dependent variable of the regression, which is the log difference over seven days in the number of new confirmed cases, overshot the out-of-sample forecasts of the regression from mid-June to August 2021. The out-of-sample forecasts overpredicted the dependent variable for the rest of 2021. The regression of the out-of-sample forecast errors on the Delta-variant share of new confirmed cases and the twice-vaccinated share of the population implies that even though the out-of-sample forecast errors can be fitted to a linear combination of the two explanatory variables up to mid-June 2021 and from August 2021 onward, significant positive forecast errors (i.e., the overshooting of the rate of change in the number of new confirmed cases) from mid-June to July 2021 are left unexplained.

## Contribution of the Go-To-Travel campaign to the spread of COVID-19 in Japan

There was controversy over the Go-To-Travel campaign in Japan between late July and late December 2020 regarding its contribution to the spread of COVID-19 in the country [61]. Fig 10 confirms a non-negligible contribution from an increase in household expenditures on domestic travel packages to an increase in the rate of change in the number of new confirmed cases during the campaign period. This result complements the cross-sectional study that confirmed an association between participation in the Go-To-Travel campaign and the incidence of symptoms indicative of COVID-19, using Internet survey data [28]. At the same time, Fig 10 also shows that the spread of COVID-19 during this period was not entirely due to the Go-To-Travel campaign, as there were contributions from other components of household expenditures as well.

A caveat is that some part of the increase in household expenditures on domestic travel packages during the campaign period might have happened even without the Go-To-Travel campaign. Therefore, to measure the campaign's contribution to the spread of COVID-19 precisely, it is necessary to identify counterfactual demand for domestic travel packages when there is no implementation of a Go-To-Travel campaign. This issue needs to be addressed in future studies.

## Limitations and strengths

The estimated regression coefficients of household expenditures, shown in Table 2, are measured per the real value of spending per household. Thus, they compare the economic value of each type of consumer activity to its contribution to the spread of COVID-19. These estimates complement studies that used mobility data to estimate the associations between people's physical visits to retail services and the spread of COVID-19 [29–34]. In particular, [29] found

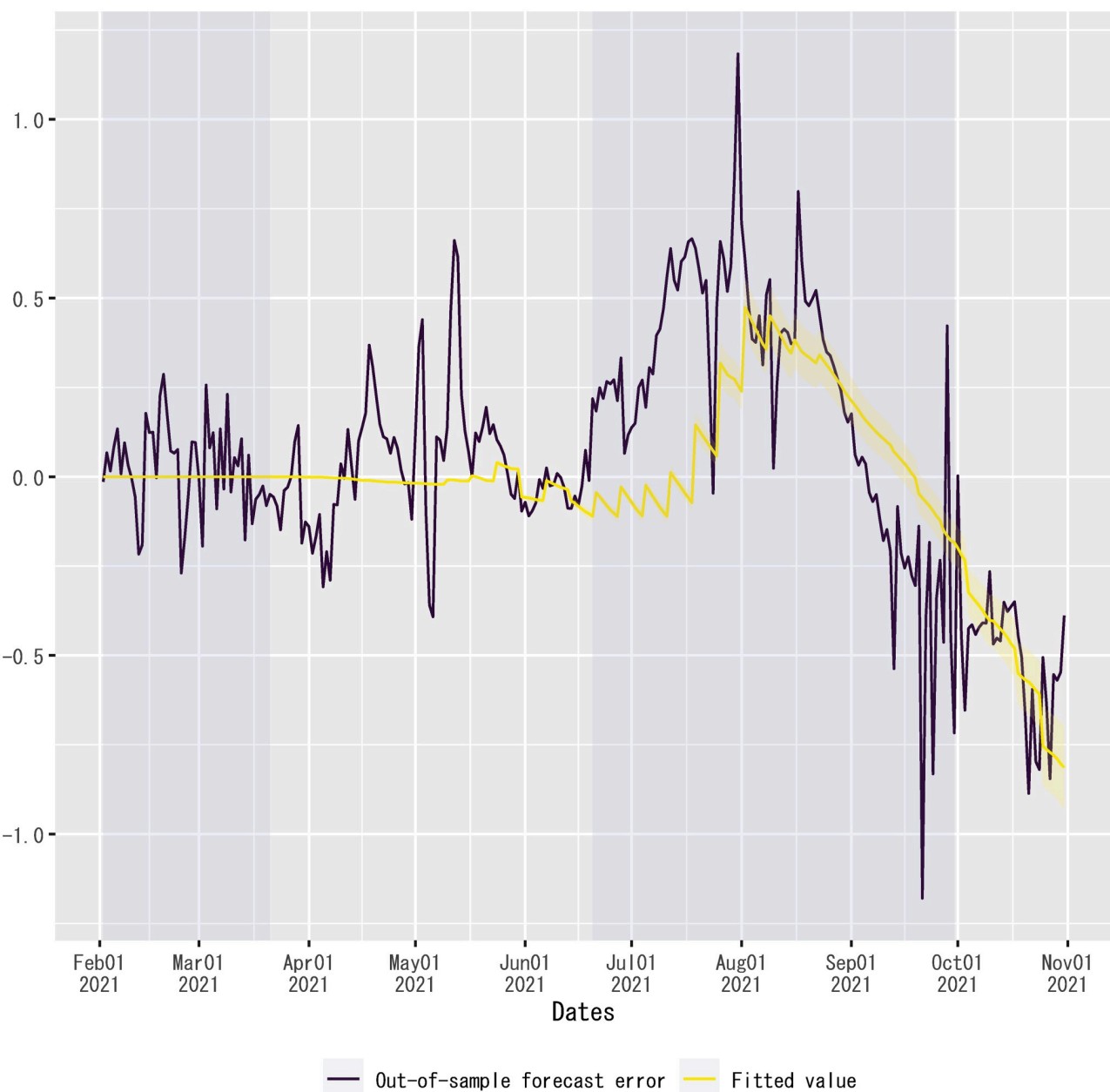

**Fig 13. Fitted values of the regression of out-of-sample forecast errors.** Notes: "Out-of-sample forecast errors" are out-of-sample forecast errors generated by the regression model with the posterior means of regression coefficients reported in S2 Table "Fitted values" are the fitted values of the regression of the out-of-sample forecast errors reported in Table 4.

that visits to full-service restaurants, fitness centers, cafes and snack bars, hotels and motels, and limited-service restaurants contributed more to the spread of COVID-19 than other types of retail services. The estimated regression coefficients shown in Table 2 have a similar implication for the rankings of infectiousness among the classified components of household expenditures. The present study also has a unique strength over studies using mobility data, as mobility data do not contain information about the amount of money spent on each retail

service visited by mobile phone users. In contrast, the household expenditure data used in the present study include such information.

Furthermore, the estimate of infectiousness of each type of consumer activity per the real value of spending may be useful when simulating how much value of consumer demand for each type must be suppressed to keep the number of new confirmed cases from rising. Table 2 indicates that among the classified components of household expenditures included in the explanatory variables, it is the most cost-effective to restrict alcoholic drinks at bars and restaurants, followed by soft drinks, confectioneries, and fruits, because reducing the real value of spending on these retail services by one unit would decrease the rate of change in the number of new confirmed cases more than reducing the real value of spending on the other components of household expenditures by one unit.

A caveat is that the estimated regression coefficients may reflect not only the direct effect of each classified component of household expenditures, but also the indirect effects of associated consumer activities summed into "the other household consumption expenditures" among the explanatory variables. They may also include the effects of workplace activities for each consumer service. To disentangle the effects of each type of retail service and the effects of workplace activities precisely, mobility data are likely to have an advantage over household expenditure data. Therefore, the present study is not superior but complementary to studies using mobility data.

It is also important to note that households adjust the composition of their consumption endogenously if the government restricts consumer activities. Therefore, this adjustment must be considered when setting a scenario on household expenditures for the simulations of the spread of COVID-19 with government interventions in the future. However, this issue is beyond the scope of the present study and is left to future research.

The estimation results reported in this study can also be useful in simulating seasonal fluctuations in the spread of COVID-19. The decomposition of out-of-sample forecasts of the regression shown in Figs 7 and 8 demonstrates that idiosyncratic fluctuations in each component of household expenditures contributed to the rate of change in the number of new confirmed cases of COVID-19. These idiosyncratic fluctuations were not only due to government interventions in retail services during each state of emergency but also due to seasonal demand, such as farewell and welcome parties with colleagues and schoolmates around the end of March, which is the end of the fiscal year in Japan, and eating out and traveling in mid-May and mid-August, which are holiday periods called the Golden Week and the Obon, respectively, in Japan. Thus, the estimates of associations between the classified components of household expenditures and the spread of COVID-19 reported in this study may be useful for simulating seasonal fluctuations in the rate of change in the number of new confirmed cases on future dates. To confirm the seasonal patterns of household expenditures, see S5 Fig, which plots each classified component of household expenditures per household from 2019 to 2021.

However, a challenge for such a simulation is to add the net effect of mutant strains and vaccinations to the regression model. As demonstrated by the regression results reported in Table 4, one possible way to identify the net effect is to regress the out-of-sample forecast errors of the regression on each mutant strain's share of new confirmed cases and the vaccinated share of the population for each number of vaccinations. This work would require consideration of the decay in the protective effect of vaccination and the presence of multiple mutant strains. Further investigation into this issue remains a challenge for future research.

Another limitation of this study is the use of nationwide data. The contribution of household expenditures to the spread of COVID-19 may differ across regions in Japan. This study uses nationwide data because government statistics on regional household expenditures report only monthly averages [45].

Therefore, the time-series regression model in this study cannot be applied to the rate of change in the number of new confirmed cases in each region because it would not be able to capture cross-regional effects such that an effective contact in one region caused an infection in another region through people's travel. To capture such an effect, it is necessary to estimate the local infectiousness of infected individuals in each region, such as the regional time-varying reproduction number, and then evaluate the associations between fluctuations in the measure of local infectiousness of infected individuals and the components of household expenditures in the region. To the best of our knowledge, this type of regional analysis has been conducted only with mobility data in Japan [35, 36]. This issue is important because the local infectiousness of household expenditures is likely to depend on population density, which varies substantially across regions in Japan. In addition, even though the regression model in this study shows a similar out-of-sample forecast performance regardless of whether it includes a weather variable among the explanatory variables or whether it includes absolute humidity or outside temperature as a weather variable, exploiting regional differences in weather conditions may help to identify the effects of weather conditions on COVID-19 incidence and transmission more precisely than in the present study. Therefore, integrating household expenditure data into a regional analysis warrants further research.

In addition, a possible alternative to the present study is to estimate a regression model with structural breaks to capture the effects of the spread of mutant strains and the progress of vaccinations using all available sample periods. On the one hand, the present study has a strength over this alternative, because it can avoid the possibility of misspecification of the effects of mutant strains and vaccinations by estimating a regression model only for the sample period before the spread of mutant strains and the progress of vaccinations. Moreover, this approach allows using the remaining sample period as testing data for model evaluation. On the other hand, the present study cannot be precise regarding the dates of possible structural breaks. Thus, the two approaches are complementary to each other. It remains a challenge for future research to estimate a regression model with structural breaks on the associations between household expenditures and the spread of COVID-19.

Lastly, despite a good fit of out-of-sample forecasts of the regression model in the present study up to mid-June 2021, the dependent variable of the regression significantly exceeded the fitted values of the regression during the summer of 2020 (see Fig 6), as well as the out-of-sample forecasts of the regression from mid-June to July 2021, even after considering the possible roles of the spread of Delta variant and the progress of vaccinations in causing out-of-sample forecast errors (see Fig 13). These results may be due to a temporary change in regression coefficients in summer for some seasonal reason or an additional independent factor, such as increased cross-border mobility during the Tokyo 2020 Olympic Games. Regarding the former possibility, regression coefficients will be time-varying if single-person households show significantly different seasonal consumer behavior than households with two or more persons. However, the present study does not include expenditures by single-person households among the explanatory variables due to the lack of daily data. Further investigation into these issues is warranted.

## Conclusions

By applying a time-series regression model to daily nationwide data in Japan, this study finds that the classified components of household expenditures and mobility in public transportation had stable associations with the log difference over seven days in the number of new confirmed cases of COVID-19 in Japan before June 2021, except for a low in-sample fit of the

regression for the summer of 2020. These associations are validated by a good fit of the out-of-sample regression forecasts for February and June 2021.

The estimated regression coefficients measure the infectiousness of consumer activities associated with each classified component of household expenditures per the real value of spending per household. The decomposition of fitted values and out-of-sample forecasts of the regression indicates that the spread of COVID-19 is associated with idiosyncratic fluctuations in various types of consumer activities. If it is possible to adjust the estimated regression coefficients to real-time developments of mutant strains and vaccinations, then the adjusted coefficients can be potentially useful for simulating changes in the number of new confirmed cases due to household spending on retail services. Such simulations would help in designing cost-efficient government interventions into consumer activities. Given the seasonality of household spending, they would also help predict seasonal fluctuations in the number of new confirmed cases over the year.

## Supporting information

**S1 Table. Japanese names of household expenditure variables in the family income and expenditure survey and the consumer price index.**
(PDF)

**S2 Table. Posterior mean and the credible interval of each parameter in the regression model.**
(PDF)

**S1 Appendix. How to construct the consumer price index for each explanatory variable on household expenditures.**
(PDF)

**S2 Appendix. Formula to compute absolute humidity.**
(PDF)

**S3 Appendix. How to fulfill missing values in Celsius temperature and relative humidity data published by the Japan Meteorological Agency.**
(PDF)

**S4 Appendix. Sensitivity analysis with alternative weather variables for explanatory variables.**
(PDF)

**S5 Appendix. Sensitivity analysis with offline household expenditures on clothing and footwear for an alternative explanatory variable.**
(PDF)

**S1 Fig. Nominal values of classified components of household expenditures per household in Japan.**
(PDF)

**S2 Fig. Fitted values and out-of-sample forecasts of the regression with time dummies related to states of emergency.**
(PDF)

**S3 Fig. Decomposition of fitted values of the regression with time dummies related to states of emergency.**
(PDF)

**S4 Fig. Fitted values of the regression when time dummies related to states of emergency are set to zero.**
(PDF)

**S5 Fig. Comparison of the "real" values of classified components of household expenditures per household in Japan over 2019–2021.**
(PDF)

## Author Contributions

**Conceptualization:** Hajime Tomura.

**Data curation:** Hajime Tomura.

**Formal analysis:** Hajime Tomura.

**Investigation:** Hajime Tomura.

**Methodology:** Hajime Tomura.

**Validation:** Hajime Tomura.

**Visualization:** Hajime Tomura.

**Writing – original draft:** Hajime Tomura.

**Writing – review & editing:** Hajime Tomura.

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
