## [Decision Letter · Decision Letter 0]

18 Jan 2022

PONE-D-21-36475Associations between components of household expenditures and the rate of change in the number of new confirmed cases of COVID-19 in Japan: time-series analysisPLOS ONE

Dear Dr. Tomura,

Thank you for submitting your manuscript to PLOS ONE. After careful consideration, we feel that it has merit but does not fully meet PLOS ONE’s publication criteria as it currently stands. Therefore, we invite you to submit a revised version of the manuscript that addresses the points raised during the review process.

We look forward to receiving your revised manuscript.

Kind regards,

Shinya Tsuzuki, MD, MSc

Academic Editor

PLOS ONE

Journal Requirements:

Additional Editor Comments:

I believe both reviewers assessed the manuscript appropriately and agree with the points they raised, then please respond each of them.

Reviewers' comments:

Reviewer's Responses to Questions

**Comments to the Author**

1. Is the manuscript technically sound, and do the data support the conclusions?

Reviewer #1: Yes

Reviewer #2: Partly

2. Has the statistical analysis been performed appropriately and rigorously? 

Reviewer #1: Yes

Reviewer #2: Yes

3. Have the authors made all data underlying the findings in their manuscript fully available?

Reviewer #1: Yes

Reviewer #2: Yes

4. Is the manuscript presented in an intelligible fashion and written in standard English?

Reviewer #1: Yes

Reviewer #2: Yes

5. Review Comments to the Author

Reviewer #1: Thank you for the opportunity to review this interesting paper that discuss household expenditures and the rate of change in the number of new confirmed cases of COVID-19 in Japan and author discuss potential application for this study to simulation to set policy for COVID-19 pandemic. The main contribution of this paper from my view is that it potentially delivers reference data to set COVID-19 or future pandemic economic policy by monitoring household expenditures data and trends for optimizing pandemic economic policy which was not available. Currently this paper could contain some room for improvement from some methodological point of view. Please refer potential suggestion below for your reference.

Major points;

1. In line 137 author states that “whereas their infectiousness is decreased in the absolute humidity”. From some published papers show associations of Epidemic growth of COVID-19 was not associated with latitude and temperature from UK or other countries, while other publications to some extent negate those associations of absolute humidity and infectiousness which shows this is still open to debate. Also another published study in 2020, for example reported influence of absolute humidity, temperature and population density on COVID-19 spread and decay durations by using multivariate analysis covering multi-prefecture in Japan. Therefore, from methodological standpoints, potentially reader of this paper may also would like to see the data with 1) assumption that infectiousness is not decreased in the absolute humidity and 2) additional cross effect of i) ambient temperature, and ii) population density assessed in addition to Table 2 which is the Estimated regression coefficients of classified components of household expenditures and mobility in public transportation in Japan.

2. In this study there seem to be no assessment of effect of vaccinated cohort population in the model, while Japan, it is estimated around 29.1％ as of Sep 2021 is the elderly population and some of them are vaccinated based on priority of nationwide vaccination program in first half of 2021. The household expenditure of those eldery population seems NOT negligible portion of Japanese economy and once the eldery population received second vaccination their behavior on consumption may change. It maybe helpful to provide the plausible reason(s) for not considering the effect of vaccination program in the regression model to conduct this analysis from applied economic standpoint since there could be potential impact associated with vaccination.

3. Author concluded as “there had been stable association between components of household expenditure and the spread of COVID-19 in Japan before the Delta variant in June 2021.” It is also helpful to see the result of degree of potential impact of Delta variant on household expenditure in this study since author mentioned in the conclusion section as, “If it is possible to adjust these estimates for real-time developments of mutant strains and vaccinations, then the adjusted estimates will be useful to compare the costs and benefits of a government intervention in each type of retail service that may be implemented in the future.”

Minor points;

1. For reference 41, please consider providing Japanese MHLW reference beside CDC recommendation since this study handles Japan specific situation. Also other parts please confirm if there are Japan specific reference to use please modify.

2. From figure 1-4, please consider providing better resolution figures, currently some of the figures are not optimal resolution...

Reviewer #2: The author aims to evaluate the associations between components of household expenditures and the spread of COVID-19 in Japan. With this in mind, he estimates a regression model using the Bayesian method with a non-informative prior.

In light of the current situation, where the topic evaluated is highly relevant to reducing the omicron variant's spread, we again phase the decision to limit some economic activities. After two years of the pandemic, economies worldwide are phasing crises, and therefore, the governments are less willing to adopt measures that limit economic activities (e.g., closing restaurants, bars, and stores), which can worsen the severe effects of the crisis. However, many stakeholders believe that there is no option but to continue applying regulations limiting economic activities. Consequently, as the author states, it is indispensable to estimate the effect on the COVID transmission rate of implementing these regulations. First, it is critical to identify those economic activities whose limitations are the most cost-effective. Second, to estimate and be transparent about the impact of limiting the economic activity is necessary to be accountable to a population that has suffered the negative consequences of the disease and the economic crisis it has caused.

1. Line 83: Does the author means "asymptomatic" instead of "asymptotic"?

2. Line 106-107: Regarding the sentence "The underlying assumption for Eq (6) is that each person has more chances of physical contacts with other people if there are more consumer activities in the country"

a. What about the increase in online shopping? An adjustment of people's behavior could lead to increases in household expenditures that are not necessarily related to physical contact.

b. Is it possible to disentangle the increase in household expenditure through online services (e.g., ordering food)? If the answer is not, the author should mention the implications of a change in consumption behaviors during the pandemic (i.e., increase in online shopping) on the validity of his results and conclusions.

3. The period for the independent variable considered is from December 25th, 2020, to February 19th, 2021.

a. Was it not possible to include a structural break in the equation? At least for some of the coefficients? For example, including a break for those periods when the percentage of confirmed cases related to specific COVID variants surpasses a threshold (different thresholds can be tested).

b. Similarly, for vaccination, to include a structural break when the percentage of the population vaccinated surpasses a certain threshold.

c. Interestingly, the variation in the observed values appears to be higher during 2021 than in 2020 (Fig 6). Could this be explained by an adjustment of the families to the current situation, and therefore, a change in the consumption and expenditures patterns? If so, could it suggest that the regression coefficients vary from 2020 to 2021?

4. There are indications of the out-of-sample results of a change in the behavior after June/July 2021 (Fig 6), which coincide with the last mentioned state of emergency.

a. Are there any particular reasons why the last state of emergency could have different behavior? Are those reasons related to the economic activities mentioned in the study? For example, different economic activities were limited, or the degree of limitation varies compared to the previous state of emergency (e.g., the case in which only vaccinated people could eat in restaurants).

5. Line 178-179. The author state that "There is no data for single-person households at the daily frequency in this survey."

a. Single-person households have consumption habits that might differ from those of family households. It could be expected that a single person is more willing to go out to a restaurant or has the time to share a drink with friends. For this segment of the population to make trips on the weekends or have quick holidays could probably be more manageable, giving them the chance to take advantage of the Go-to-travel program. The author should discuss the effect of the lack of data from single-person households on the validity of its results and conclusions.

6. Table 1 shows the corresponding lag of household expenditures in days. It is possible to observe important differences depending on the expenditure category. I would be interested in a brief analysis of the potential reasons for those differences.

7. The word "anomalies" is used to define the period in which the equation estimated values do not fit well the observed values.

a. The word "anomalies" is ambiguous and assumes that the equation perfectly captures what can be considered normal behavior. This assumption ignores the previously mentioned potential changes over time (e.g., changes in consumption patterns).

b. The differences between the observed and the predicted values could be related to a problem of omitted variables, for instance, periods with particular characteristics that could increase personal contacts and/or household expenditures (e.g., elections). It would be recommended that the author mention whether he believes this could be a limitation of the work.

8. Line 370: It is unclear why the author suggests that the equation is overfitted. He should further elaborate on this point.

9. There is a lot of material in the discussion that belongs to the Result section.

a. For instance, the paragraph that starts in line 400 describes what is observed in Fig 6, and the one started in line 420 what is observed in Fig 7, Fig 8, and Fig 13. The description should appear only in the result section, and the relevant consequences of the results should be mentioned in the discussion section.

b. I would avoid including additional figures in the discussion section (i.e., Fig 11 and 12). However, if the author considered them necessary for the narrative, the figures might be better located in the appendix.

10. The Conclusion mentions a stable association between the expenditure category and the rate of change in the number of confirmed cases until June 2021. Given that the regression is estimated until Feb 2021 and that there is a period of "anomalies" highlighted by the authors, it is probably not accurate to state a stable association until June 2021.

6. PLOS authors have the option to publish the peer review history of their article (what does this mean?). If published, this will include your full peer review and any attached files.

Reviewer #1: No

Reviewer #2: No

---

## [Author Response · Author response to Decision Letter 0]

27 Feb 2022

I thank Reviewer 1 and Reviewer 2 for their valuable comments. Please find below point-by-point responses to their comments.

1. Response to Reviewer 1

> Major point 1-1) and 1-2)-i)

Following the reviewer’s comments, I have added sensitivity analysis to the revised manuscript to report the estimates of a regression without any weather variable among the explanatory variables, and regressions with indicators for ambient (outside) temperature substituting the dummy variable for absolute humidity in the explanatory variables. Please see lines 412-441 on page 14 in the revised manuscript and S4 Appendix for the results of sensitivity analysis.

In the sensitivity analysis, I report the result that the out-of-sample forecast performance of the regression model is similarly good, regardless of whether no weather variable is included in the explanatory variables, or whether the dummy variable for absolute humidity is replaced with an indicator for ambient temperature. Even though the benchmark regression with the dummy variable for absolute humidity shows a slightly better out-of-sample forecast performance than the other regressions considered in the sensitivity analysis, I conclude in the revised manuscript that, using out-of-sample forecast performance for model evaluation, the present study cannot be conclusive on the importance of weather conditions for COVID-19 incidence and transmission.

In this analysis, I do not include indicators for both ambient temperature and absolute humidity in the explanatory variables at the same time, because I do not have good information to set prior distributions of the coefficients of the two variables to avoid a multi-collinearity problem, given similar fluctuations of ambient temperature and absolute humidity. I have noted in the revised manuscript that more precise identification of the effect of weather conditions on Covid-19 incidence and transmission remains a challenge for future research. Please see the Discussion section of the revised manuscript between lines 624-629 on page 19.

> Major point 1-2)-ii)

Unfortunately, household expenditure data at daily frequency are available only at national level in the government statistics (the Family Income and Expenditure Survey [45]) that this study uses, as described in lines 609-612 on page 18. To analyze the effect of population density on COVID-19 incidence and transmission, I would need regional data to have variations of population density among samples. Following the reviewer’s comment, I have noted in the Discussion section of the revised manuscript that it is important to estimate population density on the local infectiousness of household expenditures and that this issue warrants future research. Please see lines 622-624 on page 19 for this clarification.

> Major point 2

The regression model in the present study does not include the effect of vaccinations, because it is estimated using the data for the dependent variable only up to February 1, 2021. I limit the sample period for estimation in this way in order to avoid a possible structural break due to the spread of mutant strains in 2021, as described in lines 161-176 on page 6. I use data in the subsequent sample period to compute out-of-sample forecasts of the regression model for model evaluation. Because there was no large-scale vaccination effort before February 2021 in Japan, the regression model does not include the effect of vaccinations among the explanatory variables.

Nonetheless, following the reviewer’s comment, I have added a new section (titled “Preliminary regression of out-of-sample forecast errors on the Delta-variant share of new confirmed cases and the twice-vaccinated share of the population”) after the Results section in the revised manuscript to analyze the roles of the Delta-variant share of new confirmed cases and the vaccinated share of the population in explaining the out-of-sample forecast errors of the regression model. The regression analysis described in this section implies that while the Delta-variant share of new confirmed cases and the vaccinated share of the population can explain the out-of-sample forecast errors up to mid-June 2021 and from August 2021 onward, they cannot account for a rise in the rate of change in the number of new cases from mid-June to July 2021. A caveat is that this conclusion is tentative, as the regression analysis in the new section is yet to be validated by out-of-sample forecast performance. Please see lines 465-502 on pages 15-16 for the regression analysis.

> Major point 3

Following the reviewer’s comment, I have included the Delta-variant share of new confirmed cases in Japan among the explanatory variables in the regression analysis described in the response to Major point 2 in this letter. Please see lines 465-502 on pages 15-16 for the regression analysis.

> Minor point 1

Following the reviewer’s suggestion, I cite the clinical guide published by the MHLW of the Japanese government [41] instead of the CDC recommendation regarding the range of incubation periods of COVID-19. Please see lines 153-155 on pages 5-6. 

Except the citation of the CDC guideline in this part, there is no other citation of foreign public health agencies outside Japan in the revised manuscript.

> Minor point 2

To improve the resolution of figures to the satisfactory level for the PLOS ONE journal, I use the PACE website (https://pacev2.apexcovantage.com/) to convert figures into image files, following the instruction on figures by the journal. 

 

2. Response to Reviewer 2

> Comment 1

Yes. The typo is corrected in the revised manuscript. I thank the reviewer for pointing out the error.

> Comment 2-a

Following the reviewer’s comment, I have added a clarification in the revised manuscript that, because household expenditures can be made both online and offline, the coefficient of each explanatory variable in the regression captures the average effect of the variable. Please see lines 106-111 on page 4 for the clarification.

> Comment 2-b

Following the reviewer’s comment, I have added sensitivity analysis to the revised manuscript to report the estimation result of an alternative regression in which household expenditures for clothing and footwear in the explanatory variables include only offline purchases. I do not separate online and offline purchases for the other types of household expenditures in the explanatory variables, because they are offline services except “the other household expenditures”, even if the payments are made online, while “the other household expenditures” is the residual household expenditures other than classified household expenditures included in the explanatory variables. Please see lines 442-464 on pages 14-15 and S5 appendix for the sensitivity analysis.

In the sensitivity analysis, I report that the estimation results of the alternative regression with offline household expenditures for clothing and footwear are similar to those of the benchmark regression with total (i.e., the sum of online and offline) household expenditures for clothing and footwear. For a possible reason for this result, I refer to the fact that the online share of household expenditure for clothing and footwear had been stable during the sample period after a permanent increase in early 2020. I also mention the limitation of this interpretation at the end of the sensitivity analysis.

> Comment 3-a

A regression model with a structural break has a strength in the precise identification of the date of the structural break. A challenge for this approach is that the estimation of coefficients both before and after the structural break will be biased unless the regression model is correctly specified for the entire sample period before and after the structural break. This issue is especially challenging for the present study, because it is yet difficult to specify correctly in a regression model the effects of multiple mutant strains, some of which appeared only recently, and vaccinations, whose protective effects are decaying over time and may be different for different strains. One way to mitigate this challenge is to focus on an early part of the sample period clearly before observed structural breaks, such as the spread of mutant strains and the progress of vaccinations in 2021 in Japan. The present study takes this approach. Another benefit of this approach is that the regression model can be validated by out-of-sample forecasts (i.e., testing data), because not all the available sample period is used for the estimation of the regression model. In contrast, model validation by out-of-sample forecasts is usually difficult for a regression model with a structural break, because a structural break is typically introduced into a regression model to use all the available sample period for in-sample fitting.

Thus, the present study has some benefits that a regression model with a structural break does not have, and vice versa. Following the reviewer’s comment, I mention the benefit of estimating a regression model with a structural break as an issue for future research in the Discussion section in the revised manuscript. Please see lines 631-642 on page 19 for this clarification.

> Comment 3-b

Following the reviewer’s comment, I have added a new section (titled “Preliminary regression of out-of-sample forecast errors on the Delta-variant share of new confirmed cases and the twice-vaccinated share of the population”) after the Results section in the revised manuscript to analyze the roles of the Delta-variant share of new confirmed cases and the vaccinated share of the population in explaining the out-of-sample forecast errors of the regression model. Please see lines 465-502 on pages 15-16 for this analysis.

> Comment 3-c

If regression coefficients had changed significantly between 2020 and 2021, then the out-of-sample forecasts of the regression would not fit well the realized values of the dependent variable in the out-of-sample forecast period from February 2, 2021, onward. Nonetheless, it is possible that a temporary change in regression coefficients took place during the summer of 2020, for which the in-sample fit of the regression is low. Following the reviewer’s comment, this issue is added as a limitation of the present study in the Discussion section. Please see lines 643-656 on page 19.

> Comment 4

As stated in the reviewer’s comment, it is possible that people changed their behavior during each state of emergency. To capture the effect of such a behavioral change on the infectiousness of household expenditures, a time dummy for each state of emergency during the sample period is introduced in the regression model. Please see lines 119-122 on page 4 for this clarification. Because a time dummy only captures a time-varying average, the estimated coefficient of the time dummy for each state of emergency cannot tell the reason for the estimated time effect. Also, as will be described in the response to the reviewer’s comment 8, the estimated coefficients of time dummies for the states of emergencies are likely to suffer overfitting. Given this result, I do not try to interpret the estimated values of these coefficients in the revised manuscript.

> Comment 5

Because of lack of daily data for single-person households, it cannot be known if expenditures by single-person households were just proportional to expenditures by households with two or more persons, or if including them would improve the fit of the regression model to data significantly, such as the low in-sample fit during the summer of 2020. The present study reports that the regression model with expenditures by households with two or more persons can be validated by a good fit of out-of-sample forecasts of the regression. Following the reviewer’s comment, I mention the possibility of improving the fit of the regression model by including single-person household data for an explanatory variable in the Discussion section of the revised manuscript. Please see lines 651-656 on page 19 for this clarification.

> Comment 6

The presence of a lag in the sample correlation between the log difference over 7 days in the number of new confirmed cases and each large category of household expenditures shown in Table 1 is partly due to the presence of incubation periods after infections. Because it is a sample correlation, differences in lag lengths across different categories of household expenditures can be also due to differences in unobserved activities associated with each category of household expenditures. The possible effects of indirectly associated consumer activities for each classified component of household expenditures in the explanatory variables are mentioned in the Discussion section of the revised manuscript. Please see lines 571-575 on page 18 for this description. 

In addition, trying to remove the effect of such indirect channels and extract the direct effect of each household expenditure on incidence of an infection, the regression model in the present study constrains that household expenditures in the explanatory variables can affect the dependent variable on a future date only in proportion to the exogenous distribution of incubation periods. This use of the distribution of incubation periods in the regression model is described in lines 78-86 on page 3.

> Comment 7-a 

Regarding the use of the word “anomaly” in the previous manuscript, I intended to simply mean that the regression in the manuscript fails to account for the second wave of COVID-19 infection in Japan, rather than that the regression model is perfect. Following the reviewer’s comment, the sentence referred to by this comment is removed from the revised manuscript.

> Comment 7-b

Various possibilities of improving the fit of the regression model to data by incorporating a larger set of explanatory variables are mentioned in the Discussion section in the revised manuscript. Please see responses to the reviewer’s earlier comments, and also lines 643-656 on page 19. Also, I have added a new section on the regression of the out-of-sample forecast errors on the Delta-variant share of new confirmed cases and the vaccinated share of the population, as described in the response to the reviewer’s comment 3-b.

> Comment 8

Overfitting of a regression occurs when regression coefficients estimated by the in-sample fit of the regression produce imprecise out-of-sample forecasts. Following the reviewer’s comment, I have added S2 Fig (a figure in an online appendix) to the revised manuscript to report the out-of-sample forecasts of the regression in which a time dummy for the second state of emergency is not set to zero in the out-of-sample forecast period. This figure demonstrates a poor fit of the out-of-sample forecasts of the regression, despite a good in-sample fit of the regression. Please see lines 686-687 on page 20 for confirmation.

> Comment 9-a

Following the reviewer’s comment, the subsection of the Discussion section referred to by the reviewer is moved to an earlier section, which describes the regression of out-of-sample forecast errors on the Delta-variant share of new confirmed cases and the vaccinated share of the population. Please see lines 465-502 on pages 15-16 for confirmation.

> Comment 9-b

Following the reviewer’s comment, all the figures in the Discussion section in the previous manuscript is either moved to earlier sections or online appendices. In the revised manuscript, there is no figure included in the main text of the Discussion section. Please see lines 503-656 on pages 16-19 for confirmation.

> Comment 10

Following the reviewer’s comment, I have added “except a low in-sample fit of the regression for the summer of 2020” after the expression in the conclusion criticized by the reviewer. I have also fully revised the conclusion part of the abstract to avoid using the phrase “stable association”. Please see lines 661-662 on page 19 and the abstract on page 1 for confirmation.

---

## [Decision Letter · Decision Letter 1]

16 Mar 2022

PONE-D-21-36475R1Associations between components of household expenditures and the rate of change in the number of new confirmed cases of COVID-19 in Japan: time-series analysisPLOS ONE

Dear Dr. Tomura,

Thank you for submitting your manuscript to PLOS ONE. After careful consideration, we feel that it has merit but does not fully meet PLOS ONE’s publication criteria as it currently stands. Therefore, we invite you to submit a revised version of the manuscript that addresses the points raised during the review process.

We look forward to receiving your revised manuscript.

Kind regards,

Shinya Tsuzuki, MD, MSc

Academic Editor

PLOS ONE

Journal Requirements:

Additional Editor Comments:

Both reviewers basically satisfied with the given responses, however, raised a few minor concerns.

I agree with their point then please make a few minor changes before publication.

Reviewers' comments:

Reviewer's Responses to Questions

**Comments to the Author**

1. If the authors have adequately addressed your comments raised in a previous round of review and you feel that this manuscript is now acceptable for publication, you may indicate that here to bypass the “Comments to the Author” section, enter your conflict of interest statement in the “Confidential to Editor” section, and submit your "Accept" recommendation.

Reviewer #1: (No Response)

Reviewer #2: All comments have been addressed

2. Is the manuscript technically sound, and do the data support the conclusions?

Reviewer #1: Yes

Reviewer #2: Yes

3. Has the statistical analysis been performed appropriately and rigorously? 

Reviewer #1: Yes

Reviewer #2: Yes

4. Have the authors made all data underlying the findings in their manuscript fully available?

Reviewer #1: Yes

Reviewer #2: Yes

5. Is the manuscript presented in an intelligible fashion and written in standard English?

Reviewer #1: Yes

Reviewer #2: Yes

6. Review Comments to the Author

Reviewer #1: Thank you for my opportunity to review.

When reading the revised conclusion section, I think it would be better that the author could add more leading conclusion within the conclusion section in the abstract.

Current revised version is too short and it would be better to guide the potential readers of this manuscript to conclude with what this study result demonstrated using all the data in the manuscript with regard to background and objective of this study rather than ...(validated by)....please consider making revision to the conclusion with what key points of the conclusion does your result really support and significance to medical and economic fields.

Reviewer #2: I am satisfied with the answers of the author to my previous comments and only have two additional minor comments to add:

1) The Discussion section is mainly composed of the imitations subsection of the word. Although it is very comprehensive, I would suggest changing the title of this sub-section by "limitations and strengths" since the author also highlights some of the analysis's main points. Additionally, I would recommend mentioning additional strengths of the analysis, such that not only the limitations are accentuating. The Discussion and Conclusion section could also benefit from an English native speaker correction. They do not have mistakes in words or grammar, but the text could be written more naturally.

2) The conclusion does not highlight the "so what" of the paper. For example, the relevancy of the results in terms of policy recommendations should be emphasized.

7. PLOS authors have the option to publish the peer review history of their article (what does this mean?). If published, this will include your full peer review and any attached files.

Reviewer #1: No

Reviewer #2: No

---

## [Author Response · Author response to Decision Letter 1]

21 Mar 2022

I thank Reviewer 1 and Reviewer 2 for their suggestions to improve the manuscript. Please find below point-by-point responses to their comments.

1. Response to Reviewer 1

Following the reviewer’s suggestion, the Conclusion section of the abstract is changed to a more substantial explanation of this study’s contribution: “The estimated model can be potentially useful in simulating changes in the number of new confirmed cases due to household spending on retail services, if it can be adjusted to real-time developments of mutant strains and vaccinations. Such simulations would help in designing cost-efficient government interventions.”

To accommodate the longer conclusion section within the word limit for the abstract, the Objective section of the previous abstract is dropped from the revised manuscript. Because the Objective section overlapped the Methods section in the previous abstract, there is no significant loss of information caused by this revision.

2. Response to Reviewer 2

1) Following the reviewer’s suggestion, the title of the subsection is changed from “Limitations” to “Limitations and strengths” in the revised manuscript. Please see Line 552 on page 17 for confirmation.

Furthermore, the Discussion and Conclusion section has been edited by a professional English editor at Editage, an academic editing company. The editor is a native English speaker according to the company. Most of the English editing is incorporated in the revised manuscript, except a few jargons such as “Delta variant” and “new confirmed cases”, and a few expressions that are kept to be consistent with the earlier sections of the revised manuscript. I believe the Discussion and Conclusion section has become easier to read, as the reviewer suggested.

2) Following the reviewer’s suggestion to make the Conclusion section more substantial, the end of the Conclusion section is revised to clarify that the findings reported in this study can be potentially used for designing cost-efficient government interventions in consumer activities to contain the spread of COVID-19, and also for predicting seasonal fluctuations in the number of new confirmed cases of COVID-19. Please see lines 674-681 on page 20 for confirmation. For the same purpose, the Conclusion section of the abstract is also revised, as described in the response to Reviewer 1 above.

---

## [Editor Report · Decision Letter 2]

31 Mar 2022

Associations between components of household expenditures and the rate of change in the number of new confirmed cases of COVID-19 in Japan: time-series analysis

PONE-D-21-36475R2

Dear Dr. Tomura,

We’re pleased to inform you that your manuscript has been judged scientifically suitable for publication and will be formally accepted for publication once it meets all outstanding technical requirements.

Kind regards,

Shinya Tsuzuki, MD, MSc

Academic Editor

PLOS ONE
---

## [Editor Report · Acceptance letter]

4 Apr 2022

PONE-D-21-36475R2 

Associations between components of household expenditures and the rate of change in the number of new confirmed cases of COVID-19 in Japan: time-series analysis 

Dear Dr. Tomura:

I'm pleased to inform you that your manuscript has been deemed suitable for publication in PLOS ONE. Congratulations! Your manuscript is now with our production department. 

Kind regards, 

on behalf of

Dr. Shinya Tsuzuki 

Academic Editor

PLOS ONE